



# EVALUATION OF DIFFERENT METHODS TO MODEL NEAR-SURFACE TURBULENT FLUXES FOR AN ALPINE GLACIER IN THE CARIBOO MOUNTAINS, BC, CANADA

Valentina Radić[1], Brian Menounos[2], Joseph Shea[3,4], Noel Fitzpatrick[1], Mekdes A. Tessema[1], and Stephen J. Déry[2]

[1]Earth Ocean and Atmospheric Sciences Department (EOAS), The University of British Columbia, Vancouver, Canada
[2]Natural Resources and Environmental Studies Institute and Geography Program, University of Northern British Columbia, Prince George, Canada
[3]International Centre for Integrated Mountain Development (ICIMOD), Kathmandu, Nepal
[4]Centre for Hydrology, University of Saskatchewan, Saskatoon, Canada

*Correspondence to:* Valentina Radić (vradic@eoas.ubc.ca)

**Abstract.** As part of surface energy balance models used to simulate glacier melting, choosing parameterizations to adequately estimate turbulent heat fluxes is extremely challenging. This study aims to evaluate a set of bulk methods commonly used to estimate turbulent heat fluxes for a sloped glacier surface. The methods differ in their parameterizations of the bulk exchange coefficient that relates the fluxes to the mean meteorological variables measured 2 m above a glacier surface. The performance
of 23 bulk approaches in simulating 30-min sensible and latent heat fluxes is evaluated against the measured fluxes from an open path eddy-covariance (OPEC) method. The evaluation is performed at a point scale of an alpine glacier, using one-level meteorological and OPEC observations from a multi-day periods in the 2010 and 2012 summer season. The analysis of the two independent seasons yielded similar findings, listed as following. The bulk method, with or without the commonly used Monin-Obukhov (M-O) stability functions, overestimates the turbulent heat fluxes over the observational period, mainly due to
an overestimation of the momentum flux. In the absence of OPEC-derived M-O stability parameter, no method can successfully predict this parameter, which results in poor performances of the M-O stability corrections and consequently the bulk method. The OPEC-derived 30-min momentum flux is linearly related to the measured wind speed, contrary to the proposed quadratic relation by the commonly used bulk methods. An approach based on a katabatic flow model, which assumes a linear relation between the shear stress and the wind speed, outperforms any other bulk approach that we tested in simulating the momentum
flux. In agreement with the katabatic flow model, we show that in a more stable atmosphere the bulk exchange coefficient for momentum is smaller. The sensible heat flux can be more successfully modeled if the bulk exchange coefficients for momentum and heat are allowed to follow different parametrization schemes, rather than assuming equal schemes as is the case in the common bulk methods. Further data from different glaciers are needed to investigate any universality of these findings.



# 1 Introduction

Mountain glaciers of British Columbia are experiencing significant mass losses in response to ongoing climate change and are projected to lose most of their current volume by the end of the century (Radić et al., 2014; Clarke et al., 2015). These projections, however, are highly sensitive to the representation of glacier mass balance in the models, in particular to the model parameterizations used to simulate surface melting. All regional to global scale projections of glacier mass changes rely on a relatively simple representation of glacier melting via so-called 'degree-day' or temperature-index' models (Radić and Hock, 2014). Despite relying on empiricism, these models have relatively successfully dealt with the absence of fine-resolution meteorological input data that are, on the other hand, a prerequisite for a good performance of more physics-based surface energy balance (SEB) models. However, with the ongoing rapid development of computational recourses and global climate models, it is only a matter of time for SEB models to become routinely used in the studies of glacier evolution on regional and global scales. To ensure that future models are firmly routed in theory, new methods are required to test and parameterize turbulent heat fluxes at the glacier surface.

The turbulent sensible and latent heat fluxes are recognized as important components of the SEB over mid-latitude glaciers worldwide. When averaged over a longer period (e.g. weeks, months) these fluxes are generally less than net radiation flux. Over daily to hourly time scales, however, they can exceed the net radiation (Hock, 2005). Highest melt rates at these short temporal scales often coincide with times of greatest turbulent heat flux (Hay and Fitzharris, 1988). Direct measurements of turbulent fluxes, by eddy-covariance method, are relatively rare because they require sophisticated instrumentation with continuous maintenance which makes them unsuitable for long-term operational purposes. Few studies exist that have directly measured turbulent fluxes over mountain glaciers and all of these studies are restricted to shorter time intervals, e.g. weeks to couple of months (e.g., Munro, 1989; Forrer and Rotach, 1997; van der Avoird and Duynkerke, 1999; Cullen et al., 2007; Conway and Cullen, 2013). A more common approach in the studies of SEB on glaciers is to derive the turbulent fluxes through parameterization schemes that utilize the observations of mean meteorological variables such as near-surface air temperature, wind, and relative humidity (e.g. Hock and Holmgren, 2005; Sicart et al., 2005; Mölg et al., 2008; Reijmer and Hock, 2008; Mölg et al., 2009; MacDougall and Flowers, 2011; Sicart et al., 2011; Huintjes et al., 2015; Prinz et al., 2016). The simplest and most widely applied method is the bulk aerodynamic method (bulk method) (e.g., Braithwaite et al., 1998; Oerlemans, 2000). The bulk method is an integrated form of a gradient-flux relation, based on the theoretical work of Prandtl (1934) and Lettau (1934), which in the surface boundary layer assumes constant vertical fluxes of momentum, heat and humidity and horizontal homogeneous conditions of the surface.

Despite the common usage of bulk methods in glacier SEB studies, there are many uncertainties in its application. We summarize the main limitations of bulk methods as applied to glacier SEB studies:

1. **Surface conditions**. The bulk method for sensible heat fluxes relies on the assumption that the difference between near surface air temperature, often measured at 2 m above the surface, and surface temperature well represents the near-





surface gradient of air temperature. Furthermore, the surface temperature is often not directly measured but assumed or modeled, which can introduce a significant error in the estimates of the turbulent heat fluxes (Conway and Cullen, 2013). Similarly, for the latent heat fluxes, the surface vapor pressure is assumed rather than measured.

2. **Roughness lengths of momentum ($z_{0v}$), temperature ($z_{0t}$) and humidity ($z_{0q}$).** To get accurate roughness lengths, detailed measurements at a glacier site are needed, either by an eddy-covariance method or by vertical profile measurements. Without them, roughness lengths from other studies are used assuming that the values are universally applicable on similar glacier surfaces (e.g., Greuell and Oerlemans, 1989; Konzelmann and Braithwaite, 1995). Values of $z_{0v}$ derived at glacier ice surface, however, vary widely in space and time (Bintanja and van den Broeke, 1995; van den Broeke, 1996). There are few directly measured values for the scalar roughness lengths ($z_{0t}$ and $z_{0q}$) at valley glaciers, which led to approaches that assume their values approximate $z_{0v}$. While this 'effective' roughness length (Braithwaite, 1995) works well as a tuning parameter when the modeled turbulent fluxes are optimized to match the observed ones, it differs from its 'actual' $z_{0v}$ which is dependent only on the geometry and distribution of the roughness elements. An alternative approach to derive scalar roughness lengths relates them to measured $z_{0v}$ via a surface renewal theory of Andreas (1987). This approach found that $z_{0t}$ and $z_{0q}$ over a glacier surface are approximately two orders of magnitude smaller than $z_{0v}$ in an aerodynamically rough flow (Smeets et al., 1998; Conway and Cullen, 2013), but more studies are needed to investigate whether this finding is invariant in space and time. When the eddy-covariance method is used, accurate roughness lengths can only be confidently derived in near-neutral stability conditions that are rarely present at sloped glacier surfaces dominated, during a melting season, by a stable atmospheric stratification often accompanied by a downslope drainage (katabatic) flow. Better constrained values of roughness lengths are needed, especially considering that order of magnitude changes in $z_{0v}$ and $z_{0t}$ alters the modeled turbulent heat fluxes by a factor of two (Munro, 1989; Hock and Holmgren, 1996).

3. **Stability corrections.** Many SEB models applied to snow and ice assume that logarithmic vertical profiles of wind speed, temperature and humidity are valid under prevailing stable conditions and no modification due to the stability effects are necessary (e.g., Munro, 1991; Oerlemans, 2000; Machguth et al., 2006). Experimental and theoretical evidence supports the assertion that Monin-Obukhov (M-O) theory, which requires modification of the fluxes due to the stability effects, is not applicable over sloping glacier surface due to violation of the assumptions such as homogeneous, infinite, flat terrain and constant fluxes with height (Denby and Greuell, 2000). When the atmospheric stability is considered in the bulk method, two schemes are commonly used: (a) the approach that accounts for stability through the bulk Richardson number ($R_{ib}$) (e.g., Wagnon et al., 2003; Mölg et al., 2008; Anderson et al., 2010; Gillett and Cullen, 2011), and (b) the approach that uses a M-O stability parameter ($\frac{z}{L}$) (e.g., Braithwaite, 1995; Klok et al., 2005; van den Broeke et al., 2005; Hulth et al., 2010). In both schemes, the stability corrections are empirical functions often derived from detailed field studies on a flat and vegetated terrain. Applying the corrections through the two metrics ($R_{ib}$ and $\frac{z}{L}$) can significantly alter the modeled heat fluxes, especially in the presence of large positive near-surface temperature gradient during conditions of low wind speed ($< 3 \text{ m s}^{-1}$) (Conway and Cullen, 2013).



4. **Stable boundary layers accompanied by katabatic flow**. Over a sloping glacier surface, a low-level katabatic jet is a common feature, which means that buoyancy enters the horizontal momentum equation and the turbulence arises independently of the surface roughness (e.g., van der Avoird and Duynkerke, 1999). The Prandtl model for katabatic flow that treats eddy viscosity as a constant value (Prandtl, 1942) is unable to correctly describe the sharp near-surface gradients in wind speed and air temperature that are often observed (Munro, 1989; Oerlemans, 1998). Grisogono and Oerlemans (2001) showed that the Prandtl model can be improved if a varying assigned eddy viscosity profile is used instead of a constant value. Using this 'enhanced' Prandtl model, Parmhed et al. (2004) found that modeled turbulent fluxes at a glacier surface compare well with the measured fluxes during katabatic flows. An alternative approach to incorporate the model for katabatic flow into the bulk method for turbulent heat fluxes was developed following the classical Prandtl model for slope flows (Oerlemans and Grisogono, 2002). The model is simplistic and therefore agrees only in the overall pattern that shows an increase in sensible heat flux in response to an increase in air-surface temperature difference. Despite its simplicity, application of the model requires calibration with detailed observations of temperature and wind profiles in the surface boundary layer, in addition to the near-surface meteorological observations. Few such measurements exist on glaciers and therefore these models have not been rigorously evaluated so far.

In the light of the above, further research is needed to evaluate and develop methods that are suitable for determining near-surface turbulent fluxes over a sloping glacier surfaces subject to katabatic flows. In particular, our objective is to evaluate the existing framework of bulk approaches commonly used for parameterizing the turbulent momentum and heat fluxes at a glacier surface. We achieve this by comparing the modeled fluxes, i.e. output of the bulk approaches, to measured fluxes, i.e. the fluxes derived from an open-path eddy-covariance (OPEC) method. OPEC measurements are obtained from an alpine glacier in British Columbia (BC), over a short-term window (weeks) for two summer seasons. In the sections to follow we start with a brief overview of the field site, data collection, eddy-covariance data treatment and overall methodology. This will be followed with a detailed description of the bulk approaches and their evaluation results, and finalized with the discussion and conclusions.

## 2 Data and Methods

### 2.1 Study site and station setup

The study uses data collected from Castle Creek Glacier, an alpine glacier in the Cariboo Mountains, BC (Fig. 1). This 9.5 km$^2$ mountain glacier flows north for 5.9 km, has an elevation range of 2827-1810 m a.s.l. and contributes meltwater to Castle Creek, a tributary of the Fraser River. The glacier's annual mass balance is monitored since 2009 (Beedle et al., 2014). We utilized the data from an automatic weather station on a glacier (AWS$_{glac}$): in summer 2010, the AWS$_{glac}$ was installed at 53° 3' 2.99" N and 120° 26' 39.57" W and an altitude of 1967 m a.s.l, while in summer 2012 the AWS$_{glac}$ was installed within ± 10 m of the chosen position in 2010. In the glacier vicinity, at the lateral and terminal moraines, two year-round automatic weather stations have been in operation since 2007/2008 (Déry et al., 2010). The stations are referred to as AWS$_{up}$ (53° 2' 36"





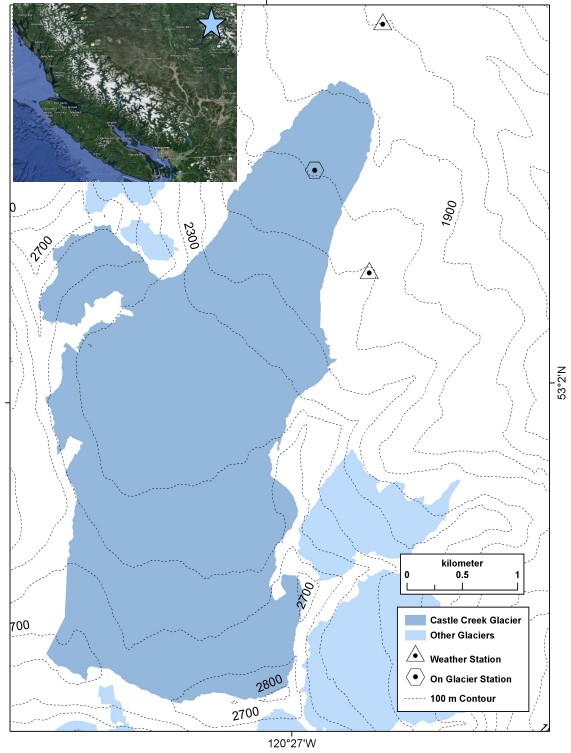

**Figure 1.** Map of Castle Creek Glacier with locations of automatic weather station on the glacier (AWS$_{glac}$) and two AWS in the glacier vicinity.

N, 120° 26' 18" W, 2105 m a.s.l.) and AWS$_{low}$ (53° 3' 45" N, 120° 26" 4" W, 1803 m a.s.l.). At AWS$_{up}$, the mean annual air temperature over 2007-2010 was -2.6°C, while summer (Jun-Aug) mean air temperature was 6.6°C. Over the same period, precipitation during summer (Jun-Aug) averages 94 mm at AWS$_{up}$ although rainfall is likely underestimated due to gauge undercatch at the exposed ridge site where mean monthly wind speeds often exceed 5 m s$^{-1}$ (Déry et al., 2010). The lower

5    part of the glacier (< 2100 m a.s.l.), where the AWS$_{glac}$ is located, is gently sloping with an approximate mean gradient of 7°.

     The AWS$_{glac}$ recorded data over a 12-day period (1-12 Aug) in 2010, and a 29-day period (21 Aug - 18 Sep) in 2012. Meteorological data were sampled at 10-second increments, and 10-minute averages were recorded with a Campbell Scientific CR1000 datalogger. Temperature and relative humidity were measured at a height $z_t$=1.7 m above the ice surface with a

10    Rotronic T/RH sensor, while wind speed and direction were measured at $z_v$=2.0 m (2010 season) and $z_v$=1.9 m (2012 season) with a RM Young wind monitor. In addition to these variables, radiation fluxes (incoming and reflected shortwave-, incoming longwave-, and net radiation) were also recorded. Surface air pressure and liquid precipitation were recorded at the AWS$_{up}$ and AWS$_{down}$ and the measured surface air pressure values were linearly interpolated to represent the surface air pressure at




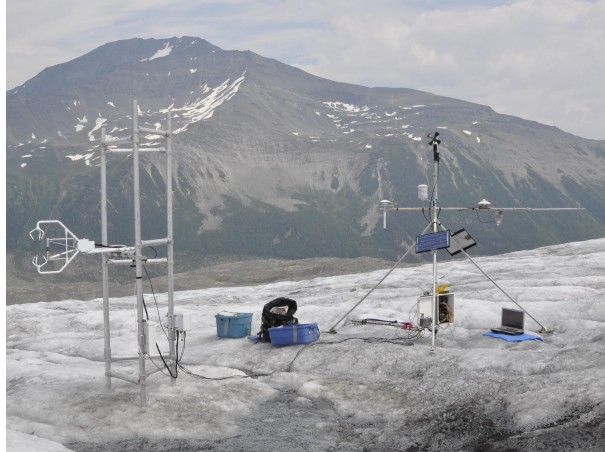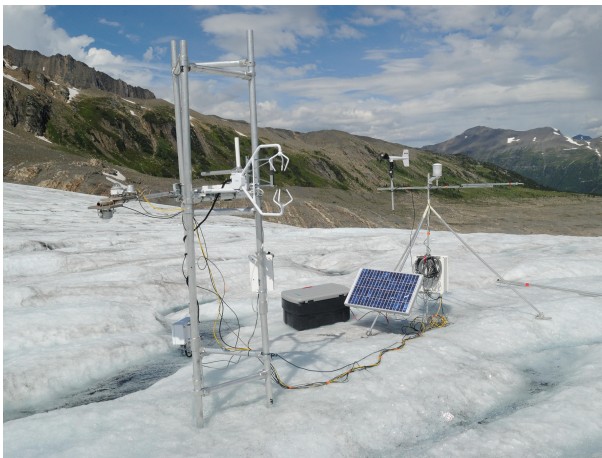

**Figure 2.** $AWS_{glac}$ with eddy covariance tower and floating meteorological tripod during installation, Castle Creek Glacier, (a) 2010 (photo by T. Mlynowski), (b) 2012 (photo by V. Radić).

$AWS_{glac}$. The meteorological sensors at $AWS_{glac}$, mounted on a floating tripod (Fig. 2), maintained a constant measurement height above the surface over the observational period. At a 3 m distance from the floating tripod, a Campbell Scientific open-path eddy covariance (OPEC) system and sonic ranger (SR50) were installed on a separate tower drilled into the ice (Fig. 2) (Jarosch et al., 2011). As part of the OPEC system, a sonic anemometer (CSAT-3) and a krypton hygrometer (KH2O) were

mounted with an initial height of 1.86 m (2010 season) and 2.0 m (2012 season) above the surface.

## 2.2  Eddy-covariance data processing

We applied a series of data processing steps as recommended by Burba (2013) that are part of the Eddy-Pro Software used to process the OPEC data and calculate the turbulent fluxes. The Eddy-Pro processing output yields turbulence statistics and calculated fluxes for each 30-min segment of the raw data. The height of OPEC sensors relative to the ice surface changed

due to the surface melting throughout the observational period: the net change of 0.70 m over 12 days in 2010, and 0.75 m over 29 days in 2012. We applied the following corrections to the raw EC data and, as later tested, none of them accounting for differences in the sensor's height produced substantial (>1 %) differences in 30-min averaged turbulent fluxes. Three-dimensional (3-D) wind speed, sonic temperature and vapor density fluctuations were recorded at 20 Hz on the datalogger (vapor density fluctuations were later converted to specific humidity fluctuations). At the beginning of each observational

season we leveled the instruments to horizontal, and have recored a minimal tilt at the end of each observational period. In both seasons, we set CSAT-3 so that the sensor's arm formed a 90° angle to the prevailing down-glacier wind direction to minimize flow distortion due to air flow through the station structure. Before calculating the turbulence statistics, we applied a coordinate rotation to the velocity data to align the streamlines into the mean flow using a planar fit method (Wilczak et al., 2001). This method sets the vertical axis of the CSAT-3 perpendicular to a hypothetical plane produced from averaged wind measurements

during the observation period. Because the glacier surface undergoes changes with time, we applied sensitivity tests to assess




whether the alignment of the plane needed adjustment and whether a different coordinate rotation method (e.g. double or triple rotation scheme) produced different output. We recored no substantial changes (>1 % difference) in the 30-min turbulent fluxes in these tests. The latent heat flux data were corrected for oxygen absorption by the KH2O (Tanner et al., 1993) and for the fluctuations in the water vapor density measurements not caused by turbulent eddies (Webb-Pearman-Leuning correction; Webb et al., 1980). Finally, we applied a series of standardized processing steps consisting of corrections for potentially high and low frequency loss, and sensor separation (Ibrom et al., 2007; Moncrieff et al., 2004; Horst and Lenschow, 2009).

### 2.3 The algorithm and quality control for roughness lengths

The covariances of the 3-D wind velocities, temperature and specific humidity, derived by the OPEC system for each 30-min data segment, were transfered to the friction velocity ($u_*$), temperature scale ($\theta_*$) and specific humidity scale ($q_*$) using:

$$u_* = \left( \overline{u'w'}^2 + \overline{v'w'}^2 \right)^{\frac{1}{4}}, \tag{1}$$

$$\theta_* = -\frac{\overline{w'T'}}{u_*}, \tag{2}$$

$$q_* = -\frac{\overline{w'q'}}{u_*}, \tag{3}$$

where $u'$ and $v'$ are fluctuations of the horizontal wind component around their 30-min mean values, $w'$ is fluctuation of the vertical wind component, and $T'$ and $q'$ are the temperature and specific humidity fluctuations. The OPEC system was also used to indirectly measure the Obukhov length ($L$):

$$L = -\frac{T_v \, u_*^3}{g \, \kappa \, \overline{w'T_v'}}, \tag{4}$$

where $T_v$ is the 30-min averaged virtual temperature (K), $g$ is the acceleration due to gravity (9.81 m s$^{-2}$), and $\kappa$ (0.40) is the von Kármán constant. Following Cullen et al. (2007), the turbulent scales and $L$ were then used to derive roughness lengths for each 30-min data segment:

$$z_{0v} = \exp\left[ -\kappa \, \frac{U_{z_v}}{u_*} - \Psi_v\left(\frac{z_v}{L}\right) \right] z_v, \tag{5}$$

$$z_{0t} = \exp\left[ -\kappa \, \frac{T_{z_t} - T_0}{\theta_*} - \Psi_t\left(\frac{z_t}{L}\right) \right] z_t, \tag{6}$$

$$z_{0q} = \exp\left[ -\kappa \, \frac{q_{z_t} - q_0}{q_*} - \Psi_q\left(\frac{z_t}{L}\right) \right] z_t, \tag{7}$$

where $U_{z_v}$, $T_{z_t}$ and $q_{z_t}$ are 30-min averages for the wind speed (m s$^{-1}$), air temperature (°C), and specific humidity (kg kg$^{-1}$) respectively, determined from AWS$_{glac}$ measurements at the senors' heights ($z_{v,t}$). In the absence of direct measurements, the surface temperature ($T_0$) was assumed to be at melting point (0°C) and the surface vapor pressure at saturation (6.13 hPa). The assumption of consistent melting is corroborated with the sonic ranger measurements showing persistent surface lowering throughout the observational period. In general, assuming that $T_0 = 0$°C works well on temperate glaciers during a melting season, and is more accurate than estimating the surface temperature from the longwave radiation measurements (Fairall et al.,





1998) or from a SEB closure (Hock, 2005). We nevertheless quantify errors in our results due to assumed rather than measured surface conditions. Vapor pressure ($e$) was converted to specific humidity using $q = 0.622\frac{e}{p}$, where $p$ is observed air pressure (hPa). $\Psi_v(\frac{z_v}{L})$, $\Psi_t(\frac{z_t}{L})$ and $\Psi_q(\frac{z_t}{L})$ (where $\Psi_t(\frac{z_t}{L}) = \Psi_q(\frac{z_t}{L})$) are integrated forms of universal functions based on M-O theory. Many such functions have been developed, and the most widely used is the Dyer-Businger flux-profile relationship (Businger

et al., 1971; Dyer, 1974). Following Conway and Cullen (2013), for stable stratification ($\frac{z_{v,t}}{L} > 0$) we use the expressions of Holtslag and de Bruin (1988) in the following forms:

$$-\Psi_v(\frac{z_v}{L}) = \frac{z_v}{L} + b\Big(\frac{z_v}{L} - \frac{c}{d}\Big)\exp\Big(-d\frac{z_v}{L}\Big) + \frac{bc}{d}, \tag{8}$$

$$-\Psi_{t,q}(\frac{z_t}{L}) = \Big(a + b\frac{z_t}{L}\Big)^{\frac{1}{b}} + b\Big(\frac{z_t}{L} - \frac{c}{d}\Big)\exp\Big(-d\frac{z_t}{L}\Big) + \frac{bc}{d} - a, \tag{9}$$

where $a = 1$, $b = \frac{2}{3}$, $c = 5$, and $d = 0.35$. For unstable stratification ($\frac{z_{v,t}}{L} < 0$), which was rarely present during our observa-
tional periods, we use the expressions of Dyer (1974).

Turbulence flux data from OPEC systems often contains spurious measurements. The roughness lengths derived from Eq. (5) - (7) widely vary because they rely on several mean and turbulence variables. One way to reduce this scatter is to limit analysis on high-quality data (Andreas et al., 2010). In this study we applied a series of filters to the 30-min data segments
(points) obtained during the study period following the approach of Conway and Cullen (2013) and Li et al. (2016). Our filtering algorithm proceeds as follows:

(a) 'Basic' filter for $z_{0t}$ and $z_{0v}$: removed the points for which $\frac{T_{z_t} - T_0}{\theta_*} < 0$ and $\frac{q_{z_t} - q_0}{q_*} < 0$, which produce unrealistically large roughness lengths since the exponents in Eq. (5) - (7) become much larger than unity. One of the main reasons for obtaining the negative values is that the bulk approach (finite differences) is a first-order approximation of the gradient-
flux approach, and the surface values for $T_0$ and $q_0$ are assumed and kept constant in time rather than measured directly for each 30-min segment.

(b) 'Stationarity' filter: only steady-state runs were used following the method of Foken (2008) which examines the fluxes for different averaging times. Steady-state conditions can be assumed if 5-min and 30-min averaged fluxes do not differ by more than 30 %.

(c) 'Neutrality' filter: only runs with near-neutral conditions ($|\frac{z_{v,t}}{L}| < 0.1$) were selected so that the choice of correction functions ($\Psi_v$, $\Psi_t$) is not important since $\Psi_{v,t} \rightarrow 0$ for $\frac{z_{v,t}}{L} \rightarrow 0$.

(d) 'Wind direction' filter: wind direction was restricted to a $\pm 45°$ sector around the glacier center line to minimize sensor arm interference and ensure the longest on-glacier fetch.

(e) 'Wind speed' and '$u_*$' filters: runs were selected for $U_{z_v} > 3$ m s$^{-1}$ and $u_* > 0.1$ m s$^{-1}$ as errors in deriving the
roughness lengths become comparatively large for low wind speeds and small friction velocities. In addition, accounting for a relatively strong air flow ($U_{z_v} > 3$ m s$^{-1}$) helps reduce potential errors in $T_{z_t}$ due to radiative heating of temperature sensor during periods of strong solar radiation (Huwald et al., 2009).





(f) 'Temperature' gradient' filter for $z_{0t}$: to ensure that a sufficiently large temperature gradient is detected between the surface and measurement height, we selected the runs for which $T_{z_t} > 1°$C.

(g) 'Moisture gradient' filter for $z_{0q}$: similarly as above, to ensure for sufficiently large moisture gradients between the surface and measurement height, we selected the points that satisfy $|e_{z_t} - e_0| > 0.66$ hPa, corresponding to a difference between $T_{z_t} = 0°$C and $T_{z_t} = 1°$C.

(h) 'Small values' filter applied to $z_{0t}$ and $z_{0q}$: following Andreas et al. (2010) we assumed that the surface exchange of heat and moisture cannot occur at scales smaller than $10^{-7}$ m. This scale is an approximate mean free path of air molecules at sea level. Considering that this is the minimal scale at which molecular diffusion takes place at the surface, $z_{0t}$ and $z_{0q}$ smaller than this are treated as unrealistic (i.e. eddy diffusivity can take place only at larger scales).

(i) 'Large values' filter: no study of turbulent fluxes over glaciers determined roughness lengths larger than 1 m, thus we treated any values of roughness length that exceed 1 m as unrealistic for our site.

We note that it is common to exclude the data during precipitation events as these greatly increase measurement errors associated with the OPEC system. Nevertheless, this filter was redundant in our study because the above filters already removed the points overlapping with the precipitation events (recorded at $AWS_{low}$), and because OPEC system (in particular KH20) failed to take measurements during most of these events.

## 2.4 Bulk methods

All bulk methods used in this study are rooted in the gradient-flux relations, in which the turbulent fluxes of momentum ($\tau$), sensible heat ($Q_H$) and latent heat ($Q_E$) in the surface boundary layer are proportional to the time averaged gradients of wind speed ($\overline{U}$), potential temperature ($\overline{\theta}$) and specific humidity ($\overline{q}$), expressed by:

$$\tau \equiv \rho_a \, u_*^2 = \rho_a \, K_M \, \frac{\partial \overline{U}}{\partial z}, \tag{10}$$

$$Q_H = \rho_a \, c_p \, K_H \, \frac{\partial \overline{\theta}}{\partial z}, \tag{11}$$

$$Q_E = \rho_a \, L_v \, K_E \, \frac{\partial \overline{q}}{\partial z}, \tag{12}$$

where $\rho_a$ is the air density, $c_p$ is the specific heat of air at constant pressure (1005 J kg$^{-1}$K$^{-1}$), $L_v$ is the latent heat of vaporization (2.514 MJ kg$^{-1}$), $z$ is the height above the surface, $K_M$ is the eddy viscosity, and $K_H$ and $K_E$ are the eddy diffusivities for heat and vapor exchange, respectively. Eq. (10) also defines the friction velocity $u_*$ which we use henceforth as a surrogate for surface shear stress. We note that by convention in SEB on glaciers, the heat fluxes are positive if they transport heat towards the surface, and negative if they transport heat away from the surface.



## 2.5 Methodology outline

Before we go into details, we provide an overview of the bulk approaches, whose performance in simulating the turbulent fluxes of momentum, heat and humidity, we evaluate in the study. In total we evaluate 23 bulk schemes, and the details of each scheme are provided together with the results, in order to facilitate easier readability. The modeled 30-min turbulent fluxes ($u_*$, $Q_H$, $Q_E$) are compared to the equivalent OPEC-derived fluxes with the use of standard evaluation metrics: root-mean-square-error (RMSE), mean-bias-error (MBE) and Pearson correlation coefficient (r). First, we derive 30-min turbulent fluxes for each season using the K-approach bulk method with three different parameterization schemes for the bulk exchange coefficients ($C$): (**1**) $C_{log}$ - which assumes logarithmic vertical profiles of wind, temperature and specific humidity, (**2**) $C_{Rib}$ - which assumes stability corrections to the fluxes using $R_{ib}$, and (**3**) $C_{M-O}$ - which assumes stability corrections using the universal stability functions of $\frac{z_{v,t}}{L}$. These schemes require estimates of the roughness lengths ($z_{0v,t,q}$) which we assume to be constant in time and equal to the mean log value of the OPEC-derived 30-min roughness lengths for neutral condition only (i.e. when all filters are applied to the OPEC data). In addition to these three schemes, we use $C_{M-O}$ parametrization with the scalar roughness lengths derived from the surface renewal model of Andreas (1987), yielding (**4**) $C_{SR}$ parameterization scheme. Second, we derive the turbulent fluxes using the K-approach method, but instead of modeled friction velocity ($u_*$) and modeled $\frac{z_{v,t}}{L}$ we use their OPEC-derived values. This yields the following schemes: (**5**) $C_{log}\ u_*$, (**6**) $C_{Rib}\ u_*$, (**7**) $C_{M-O}\ u_*$, (**8**) $C_{SR}\ u_*$, (**9**) $C_{M-O}\ \frac{z}{L}$, (**10**) $C_{SR}\ \frac{z}{L}$, (**11**) $C_{M-0}\ u_*\ \frac{z}{L}$, (**12**) $C_{SR}\ u_*\ \frac{z}{L}$. In the schemes with $C_{M-O}$, we apply the most commonly used integrated stability functions in glacier studies. In addition, we derive our own empirical integrated stability functions from the relations between OPEC-derived fluxes and $\frac{z_{v,t}}{L}$, which yields the parameterizations: (**13**) $C_{M-O}\ new\ \frac{z}{L}$ and (**14**) $C_{M-O}\ new$. Third, we use bulk methods derived from a simple katabatic flow model: (**15**) $C_{kat}$ method - following Oerlemans and Grisogono (2002), and an approach with an integrated varying eddy viscosity profile $K_{Int}$ (Grisogono and Oerlemans, 2001; Parmhed et al., 2004). In the $K_{Int}$ approach where we analyze whether two parameters that determine $K_{Int}$ (later introduced as $K_{max}$ and $H_K$) can be expressed as functions of OPEC-derived $\frac{z_{v,t}}{L}$. This yields two schemes: (**16**) $K_{Int}\ K_{max}\ \frac{z}{L}$ and (**17**) $K_{Int}\ H_k\ \frac{z}{L}$. Finally we test a 'hybrid' approach which combines the K-approach with the $K_{Int}$ method (with OPEC-derived or modeled $\frac{z_{v,t}}{L}$) yielding: (**18**) $C_{log}\ K_{max}\ \frac{z}{L}$, (**19**) $C_{log}\ H_K\ \frac{z}{L}$, (**20**) $C_{M-O}\ new\ K_{max}\ \frac{z}{L}$, (**21**) $C_{M-O}\ new\ H_K\ \frac{z}{L}$, (**22**) $C_{log}\ K_{max}$, and (**23**) $C_{log}\ H_K$.

## 3 Results

### 3.1 Estimates of the roughness lengths

We apply the filters discussed in Section 2.3 to the OPEC data to obtain the high-quality estimates of 30-min turbulent fluxes, which are then used to determine the roughness lengths using the Eq.(5)-(7). For the 12-day observational period in 2010, prior to the filtering, there are 515 values for $z_{0v,0t}$ while 307 values for $z_{0q}$. The fewer initial data points for $z_{0q}$ is due to high sensitivity of KH2O sensor to the build-up of droplets on the sensor window during precipitation events. During these events KH2O mainly produced spurious values or failed to record the data, while CSAT-3 recorded values, albeit with questionable

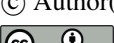



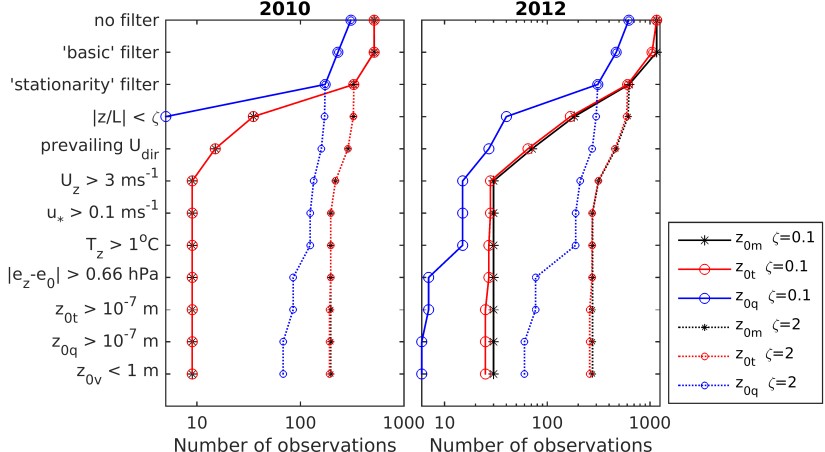

**Figure 3.** Initial number of observations for estimating the roughness lengths (from 30-min averages of OPEC data) and remaining number of observations after each filter is applied (in the order from top down) for 2010 and 2012 observational periods. Dotted line represents the same but without applying the filter for neutral stability conditions, i.e. all points that satisfy $|\frac{z_v}{L}| < 2$ are used in the calculation of roughness lengths.

quality. After filtering, usable data reduced by 98 %, giving nine high-quality estimates for $z_{0v,0t}$, but none for $z_{0q}$ (Fig. 3). Similarly, initial number of data points in the 29-day observational period in 2012 is 1161 for $z_{0v,0t}$ and 621 for $z_{0q}$, while the post-filtering number of data points are 30 for $z_{0v}$, 25 for $z_{0t}$ and six for $z_{0q}$. Among all the filters, the 'neutrality filter' ($|\frac{z_{v,t}}{L}| < 0.1$) eliminates the largest percent of the data points, followed-by the 'stationarity' filter. Despite the significant loss

of viable data, these filters are essential to achieve a more accurate determination of roughness lengths with the bulk method. OPEC-derived fluxes are assumed valid only during the steady-state or stationary conditions (i.e. in the absence of intermittent turbulence and/or gravity waves). For the neutral conditions, the choice of the stability correction functions has negligible effect on the calculated fluxes (the correction function approaches zero as $|\frac{z_{v,t}}{L}|$ approaches zero). Furthermore, the neutral stability conditions are more commonly observed during overcast and windy conditions rather than during clear-sky conditions

with prevailing katabatic flows. This fact minimizes the chance of selecting 30-min intervals with katabatic flows for which the bulk method, used to determine the roughness lengths, might be deficient. If we remove the 'neutrality' filter, and instead select all values for which $|\frac{z_{v,t}}{L}| < 2$, we obtain 196, 191, and 68 values for $z_{0v}$, $z_{0t}$, and $z_{0q}$, respectively in 2010, while for 2012 season we obtain 276, 259, and 60 values for $z_{0v}$, $z_{0t}$, and $z_{0q}$, respectively. The threshold of $\frac{z_{v,t}}{L} = 2$ is chosen because the universal stability functions for stable stratification are commonly defined up to $\frac{z}{L} = 2$ (Foken, 2008), which represents

strongly stratified stable regime.

We determine the time windows of prevailing katabatic flows (shaded areas in Fig. 4) by identifying longer periods (several hours to days) with almost stationary down-glacier wind direction (200° for our site) and relatively high mean wind speeds (>





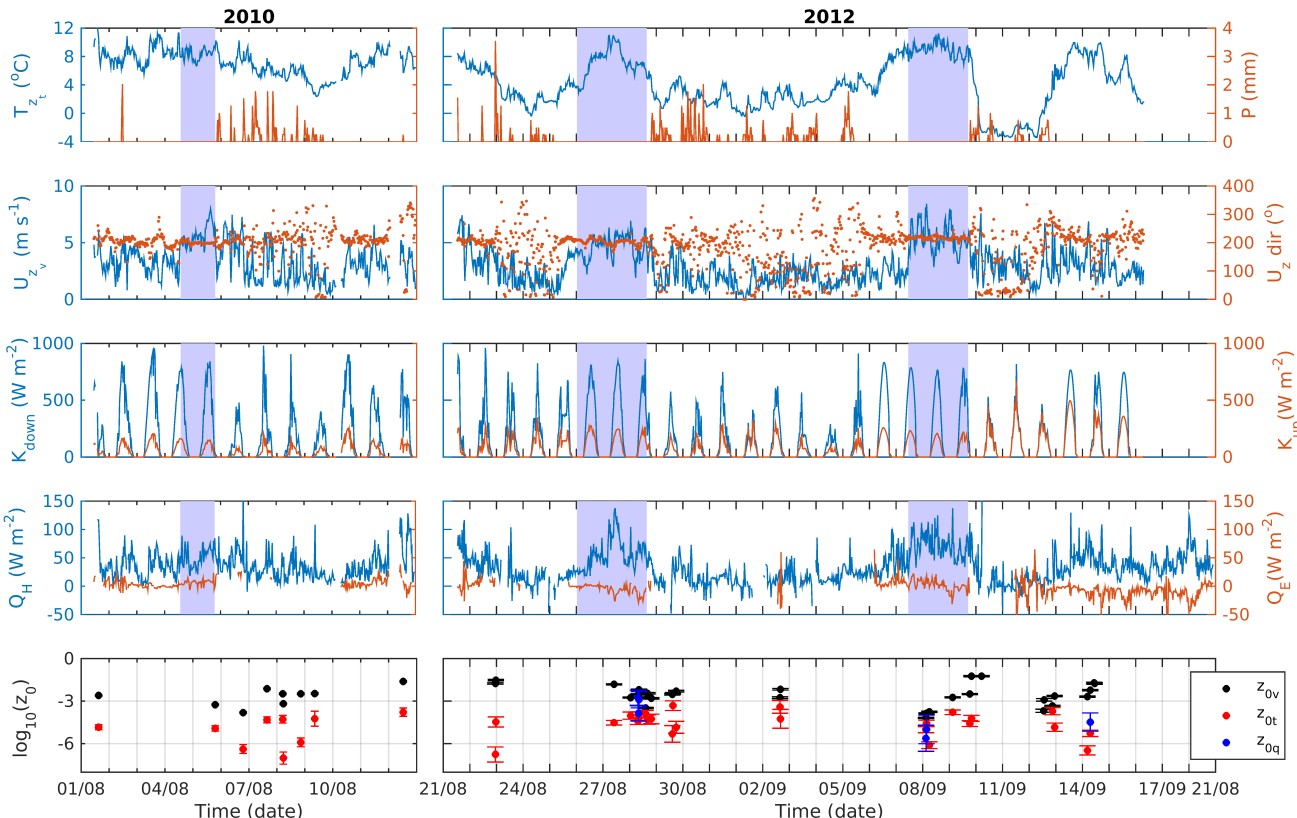

**Figure 4.** 30-min values of meteorological variables and SEB fluxes measured at the $\mathrm{AWS}_{glac}$ in 2010 (left panel) and 2012 (right panel): near surface air temperature ($T_z$), accumulated precipitation ($P$; measured at $\mathrm{AWS}_{down}$), wind speed ($U_z$), wind direction ($U_{dir}$), incoming ($K_{down}$) and reflected shortwave radiation ($K_{up}$), OPEC-derived sensible ($Q_H$) and latent ($Q_E$) heat flux. Bottom panel shows the OPEC-derived roughness lengths for momentum ($z_{0v}$), temperature ($z_{0t}$) and humidity ($z_{0q}$), after filtering OPEC data, and their estimated errors. Shaded in blue are the identified time intervals of prevailing katabatic flow (see text).

3 m s$^{-1}$), preferably reaching maximum 30-min mean values > 5 m s$^{-1}$. With this rough identification criteria we may miss some shorter time intervals with katabatic flows, but a more restrictive approach reduces the uncertainty that originates from the selection of katabatic flow periods in our further analysis. The identified prevailing katabatic periods (Fig. 4) overlap with clear sky conditions and with relatively high near-surface air temperatures ($T_{z_t}$), as is commonly expected for katabatic flows

5  at glacier surfaces (e.g., Oerlemans et al., 1999). For each roughness length estimate from 30-min segments (Eq. (5) - (7)) we also estimate its error (error bars in Fig. 4) derived through an error propagation method for a multi-variable function (e.g., Bevington, 1969), where the measurement error for wind speed and assumed errors for surface temperature and water vapor pressure are propagated (keeping just the linear terms in Taylor expansion) into the final error for the roughness length. The following error values are used: measurement error for wind speed $\delta U_{z_v}$ =0.11 ms$^{-1}$, assumed error for surface temperature

10  $\delta T_0$ =0.5°C, and assumed error for surface water vapor pressure $\delta e_0 = 0.23$ hPa. To provide an expected (mean) value for





**Table 1.** Mean and $\pm$ standard deviation ($\sigma$) of log roughness lengths of momentum ($z_{0v}$), temperature ($z_{0t}$) and humidity ($z_{0q}$) derived for neutral stability conditions ($|\frac{z}{L}| < 0.1$) during 2010 and 2012 observational period. Also shown is weighted mean $\pm$ weighted standard deviation ($\sigma_w$; see text) and number of points ($N$) used to assess these values, i.e. number of points remaining after all the filters have been applied to the original data.

| Parameter | log(mean $\pm \sigma$) | log(weighted mean $\pm \sigma_w$) | $N$ | log(mean $\pm \sigma$) | log(weighted mean $\pm \sigma_w$) | $N$ |
| :---: | :---: | :---: | :---: | :---: | :---: | :---: |
| | 2010 | 2010 | 2010 | 2012 | 2012 | 2012 |
| $z_{0v}$ | -2.67 $\pm$ 0.66 | -2.63 $\pm$ 0.67 | 9 | -2.56 $\pm$ 0.75 | -2.54 $\pm$ 0.76 | 30 |
| $z_{0t}$ | -5.08 $\pm$ 1.11 | -4.98 $\pm$ 0.98 | 9 | -4.58 $\pm$ 0.87 | -4.52 $\pm$ 0.79 | 25 |
| $z_{0q}$ | | | 0 | -4.11 $\pm$ 1.15 | -3.89 $\pm$ 1.09 | 6 |

roughness lengths during neutral conditions we took the average of their logged values given the measurement's log-normal distribution. To incorporate the individual error of each estimate into the mean value, we derive a weighted mean roughness length, where the weights are inversely proportional to the individual errors. The results (Table 1) show that the weighted mean values do not substantially differ from the mean values and we thus use the non-weighted mean values in the remaining

5 analysis. For both seasons, our results show that mean log $z_{0v}$ is of the order of $10^{-3}$ m and two orders of magnitude larger than mean log $z_{0t}$. The overall error of these mean values, expressed as one sigma ($\pm$ a standard deviation), shows uncertainty of one order of magnitude around the mean value (Table 1). To determine $z_{0q}$ for 2010 season, which had no high-quality measurements for $z_{0q}$; we thus assume $z_{0q} \approx z_{0t}$ since the equality holds for the mean log values in the 2012 season.

### 3.2 K-approach with common parameterizations for $C$

Following Conway and Cullen (2013) we use the bulk methods derived by integrating the flux-gradient equations with three different assumptions: (**1**) $C_{log}$ method - assumes that wind speed, temperature, and humidity have logarithmic profiles with height (**2**) $C_{Rib}$ method - assumed that the logarithmic profiles are modified with stability corrections based on the bulk Richardson number ($R_{ib}$), (**3**) $C_{M-O}$ method - assumes that the logarithmic profiles are modified with M-O universal stability functions of $\frac{z_{v,t}}{L}$, and (**4**) $C_{SR}$ - same as $C_{M-O}$ but with modeled scalar roughness lengths. In the following sections we

first provide a brief theoretical background and derivation of each method, which is then followed by an evaluation of those approaches.

#### 3.2.1 K-approach with logarithmic profiles for wind and temperature ($C_{log}$)

According to the mixing-length theory (Prandtl, 1934) for a neutral surface layer, eddy viscosity can be parametrized as (K-approach; Stull, 1988):

$$K_M = \kappa \, z \, u_*. \tag{13}$$



Assuming $K_M = K_H = K_E$, inserting these parameterizations in the gradient flux relations (Eq. (10) - (12)) and integrating the equations with respect to height, while treating $u_*, Q_H, Q_E$ as constants, gives the bulk aerodynamic expressions:

$$u_* = C_v \, U_{z_v},$$  (14)

$$Q_H = \rho_a \, c_p \, C_v \, C_t \, U_{z_v} \, (T_{z_t} - T_0),$$  (15)

$$Q_E = \frac{0.622}{p} \, \rho_a \, L_v \, C_v \, C_q \, U_{z_v} \, (e_{z_t} - e_0).$$  (16)

where $C_v^2 \equiv C_D$ is a dimensionless exchange coefficient ('drag coefficient') for momentum flux, while $C_v C_t \equiv C_H$ and $C_v C_q \equiv C_E$ are dimensionless exchange coefficients for the sensible and latent heat flux, respectively. Wind speed ($U_{z_v}$), air temperature ($T_{z_t}$) and water vapor pressure ($e_{z_t}$) represent the time averaged values at their measurement heights $z_{v,t}$, while zero subscript in these variables denotes their surface values (note that $z_t = z_q$). The air density at $\text{AWS}_{glac}$ ($\rho_a$) is derived as the air density at standard sea-level pressure ($\rho_0 = 1.29 \, \text{kg m}^{-3}$ at $0^\circ$C) multiplied by the ratio between the air pressure estimated at $\text{AWS}_{glac}$ ($p$) and the standard sea-level pressure ($p_0$; 1013 hPa). The dimensionless exchange coefficients for the neutral atmosphere assume logarithmic profile of wind, temperature and humidity with height, and take the following form:

$$C_{v,log} = \frac{\kappa}{\ln\left(\frac{z_v}{z_{0v}}\right)},$$  (17)

$$C_{t,log} = \frac{\kappa}{\ln\left(\frac{z_t}{z_{0t}}\right)},$$  (18)

$$C_{q,log} = \frac{\kappa}{\ln\left(\frac{z_t}{z_{0q}}\right)}.$$  (19)

### 3.2.2 K-approach with $R_{ib}$-based stability corrections ($C_{Rib}$)

The method assumes the same parameterization for $K_{M,H,E}$ (Eq. (13)) as $C_{log}$ method, but it allows a flux reduction in a stable stratification via the bulk Richardson number ($R_{ib}$) defined as:

$$R_{ib} = \frac{g \, (T_{z_t} - T_0)(z_t - z_0)}{T_{z_t} \, U_{z_v}^2},$$  (20)

where temperature is given in Kelvin. For stable conditions ($0 < R_{ib} < 0.2$) which prevail over a melting glacier surface, the coefficient $C$ for the property $y$ (either $v$, $t$ or $q$) is (Webb, 1970):

$$C_{y,R_{ib}} = \frac{\kappa}{\ln\left(\frac{z_y}{z_{0y}}\right)}(1 - 5R_{ib}).$$  (21)

### 3.2.3 K-approach with M-O stability functions ($C_{M-O}$)

Starting from the parametrization for neutral conditions (Eq. (13))) but allowing $K_{M,H,E}$ to vary in response to atmospheric stability (generally, $K$ for statically unstable surface layer > $K$ for neutral > $K$ for statically stable), $K_M$ (and equivalently $K_H$ and $K_E$) can be expressed as:

$$K_M = \frac{\kappa \, z \, u_*}{\psi_v\left(\frac{z}{L}\right)},$$  (22)





where $\psi_v$ is a dimensionless shear and $L$ is Obukhov length. Using this parametrization for $K$ and integrating the flux-gradient equations, the expression for $C$ (for the property $y$) is:

$$C_{y,M-0} = \frac{\kappa}{\ln\left(\frac{z_y}{z_{0y}}\right) - \Psi_y\left(\frac{z_y}{L}\right)}. \tag{23}$$

We use the integrated form of M-O stability functions from Holtslag and de Bruin (1988) and Dyer (1974) as expressed earlier in Section 2.3. The calculation of $\Psi$ requires an estimate of $L$, which requires an estimate of $Q_H$ and $u_*$. This thus becomes a root-finding problem for the three coupled equations, which is either solved by Newton's method or a fixed-point iteration method. We chose the latter since the method is shown to successfully work for this set of equations (e.g., Berkowicz and Prahm, 1982; Lee, 1986) and has been widely used in glacier studies (e.g., Munro, 1989; Conway and Cullen, 2013). In this iterative method each 30-min $Q_H$ and $u_*$ are initially derived assuming the neutral case ($\frac{z_{v,t}}{L} = 0$). This approach allows an estimate of $L$ and $\Psi$, which in the next iteration yields a new estimate for $Q_H$ and $u_*$. These steps are repeated until no significant change in $Q_H$ occurs (e.g. $> \pm 1$ W m$^{-2}$ difference), a condition which is satisfied within first five iterations (Munro, 1989).

### 3.2.4 K-approach with $C_{M-O}$ parameterization and surface renewal model ($C_{SR}$)

In all the parameterizations for $C$ we use a constant value for roughness lengths derived as the mean log value from the 30-min values (Eq. (5) - (7)) calculated for the neutral conditions only. An alternative approach in deriving the scalar roughness lengths ($z_{0s} \equiv z_{0t,0q}$) is through the surface renewal model by Andreas (1987). The model uses a simple similarity arguments considering the structure of the viscous sublayer to derive $z_{0s}$ from known $z_{0v}$ as a function of Reynolds roughness number ($Re_* = \frac{u_* z_{0v}}{\nu}$, where $\nu$ is the kinematic viscosity of air, equal to $1.46 \times 10^{-5}$ m$^2$ s$^{-1}$):

$$\ln\left(\frac{z_{0s}}{z_{0v}}\right) = b_0 + b_1 \ln(Re_*) + b_2 \left[\ln(Re_*)\right]^2, \tag{24}$$

where $b_{0,1,2}$ are the polynomial coefficients (Table 2). The model has been successfully tested for relatively smooth snow and ice surfaces, while for a rough, hummocky glacier surface Smeets and van den Broeke (2008) found that somewhat different values for the polynomial coefficients (Table 2) give better performance. Here we apply the model for the scalar roughness lengths (Eq. (24)) and input the modeled values for $z_{0t,0q}$ into $C_{M-O}$ method, which gives us the fourth bulk scheme in this section: (**4**) $C_{SR}$ method.

### 3.2.5 Evaluation results: K-approach with common parameterizations for $C$

We summarize the evaluation results of the bulk schemes: (**1**) $C_{log}$, (**2**) $C_{Rib}$, (**3**) $C_{M-O}$, and (**4**) $C_{SR}$. First we compare 30-min turbulent fluxes of momentum, sensible and latent heat flux, from the 2012 season, derived from the OPEC data (observed fluxes) with the modeled fluxes from each bulk scheme (Fig. 5). Only the filtered 30-min segments, except the 'neutrality' filter (Section 2.3), are used for this comparison. To estimate an error for each modeled 30-min flux (error bars in Fig. 5) we use a Monte Carlo approach where each bulk method is run 1000 times with randomly perturbed roughness length values. The roughness length values for each Monte Carlo run are picked randomly from an assumed normal distribution of their log



**Table 2.** Values for the polynomial coefficients in Eq. (24) for temperature ($b_t$) and humidity ($b_q$) for three different aerodynamic flow regimes according to (a) Andreas (1987), and (b) Smeets and van den Broeke (2008)

|  | $Re_* \leq 0.135$ | $0.135 < Re_* < 2.5$ | $0.135 \leq Re_* \leq 1000$ $z_{0v} < 10^{-3}$ m | $0.135 \leq Re_* \leq 1000$ $z_{0v} > 10^{-3}$ m |
|---|---|---|---|---|
|  | (a) | (a) | (a) | (b) |
| $b_{t0}$ | 1.250 | 0.149 | 0.317 | 1.5 |
| $b_{t1}$ |  | -0.550 | -0.565 | -0.2 |
| $b_{t2}$ |  |  | -0.183 | -0.11 |
| $b_{q0}$ | 1.610 | 0.351 | 0.396 | 1.5 |
| $b_{q1}$ |  | -0.628 | -0.512 | -0.2 |
| $b_{q2}$ |  |  | -0.180 | -0.11 |

values (Table 1). The modeled fluxes (y-axis in Fig. 5) are the mean values from the Monte Carlo ensemble, whereas the error bars are represented with $\pm$ 1 standard deviation ($\sigma$) of the ensemble. As illustrated in the scatter plots (Fig. 5), each bulk scheme overestimates the turbulent fluxes (MBE > 0) over the entire period. The scheme with the lowest RMSE for the friction velocity is $C_{Rib}$, while for the sensible and latent heat fluxes is $C_{SR}$. The variability in friction velocity (r=0.26, p-value > 0.05) is least successfully modeled, while statistically significant correlations (p < 0.05) are found for both $Q_H$ and $Q_E$ fluxes, with the highest correlation coefficients found for $C_{log}$ and $C_{Rib}$ method (Table 3). The scatter plots for the 2010 season (not shown) are similar to these for the 2012 season, while the correlation values are smaller in 2010 due to smaller sampling period (Table 3). We also calculated the total model RMSE from the individual 30-min error estimates (error bars in Fig. 5) over the entire period, and found that it is smaller than the RMSE between modeled and observed fluxes for each of the bulk methods.

As illustrated in Fig. 5, the performance of $C_{SR}$ method does not significantly differ from $C_{M-O}$ because the OPEC-derived values for scalar roughness lengths (neutral conditions only) mostly agree with the estimated values from the surface renewal model (Fig. 6). Large roughness Reynolds numbers ($Re_*$), estimated for our site, indicate a rough flow regime for which the surface renewal model (Andreas, 1987) predicts the scalar roughness lengths two orders of magnitude smaller than $z_{0v}$. This prediction agrees well with our observed values, especially when we consider the mean log values of roughness lengths rather than their individual 30-min estimates. Despite the large scatter of 30-min OPEC-derived $\frac{z_{0t}}{z_{0v}}$ values around the predicted values from the surface renewal model, more than 50 % of the observed values fall between the predictions of Andreas (1987) and those of Smeets and van den Broeke (2008) derived for hummocky ice with $z_{0v} > 1 \times 10^{-3}$ m (Fig. 6).

### 3.2.6 Evaluation results: K-approach with observed variables ($C\, u_*$, $C\, \frac{z}{L}$)

In the gradient-flux relation, used to derive the K-approach bulk methods, the eddy viscosity is parameterized as a function of $z$, $u_*$ and M-O stability parameter ($\frac{z}{L}$). If $u_*$ and $\frac{z}{L}$ are modeled rather than directly measured, the modeling error of the input

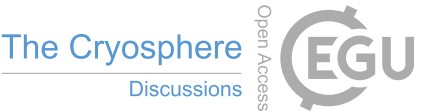

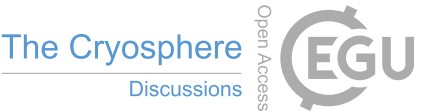

**Figure 5.** Comparison of 30-min values of observed (OPEC derived) and modeled (bulk method) shear stress (friction velocity, $u_*$), sensible ($Q_H$) and latent ($Q_E$) heat flux, using four parameterization schemes in the bulk method (see text): $C_{log}, C_{Rib}, C_{M-O}, C_{SR}$. Dots and error bars show the mean and standard deviation of Monte Carlo ensembles with $\sim$1000 simulations per point. Red points show values during the neutral conditions ($|\frac{z_{v,t}}{L}| <0.1$) only, and blue points indicate values during the prevailing katabatic flow. Root-mean-square-error (RMSE; W m$^{-2}$), mean bias error (MBE; W m$^{-2}$), and Pearson correlation coefficient (r) are provided for each scatter plot.

variables may influence the performance of the bulk methods. Our goal in this section is to investigate the influence of two components ($u_*$ and $\frac{z_{v,t}}{L}$) on the method performance. Therefore, we assess the turbulent fluxes using the (**1**)-(**4**) bulk schemes with OPEC-derived $u_*$ and OPEC-derived Obukhov length. The following methods are tested with the OPEC-derived $u_*$ only: (**5**) $C_{log}\ u_*$, (**6**) $C_{Rib}\ u_*$, (**7**) $C_{M-O}\ u_*$, (**8**) $C_{SR}\ u_*$, then with the OPEC-derived $\frac{z_{v,t}}{L}$ only: (**9**) $C_{M-O}\ \frac{z}{L}$ and (**10**) $C_{SR}\ \frac{z}{L}$, and finally with both $u_*$ and $\frac{z_{v,t}}{L}$ derived from OPEC data: (**11**) $C_{M-0}\ u_*\ \frac{z}{L}$ and (**12**) $C_{SR}\ u_*\ \frac{z}{L}$. In each scheme we use mean log roughness lengths (Table 1).



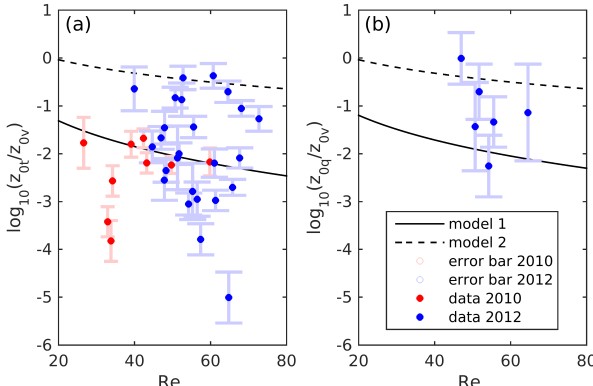

**Figure 6.** The ratio of roughness lengths for temperature ($z_{0t}$) and momentum ($z_{0v}$) (panel a) and the ratio of roughness lengths for humidity ($z_{0q}$) and $z_{0v}$ (panel b) versus roughness Reynolds number ($Re_*$). Model 1 (solid line) is the theoretical prediction of Andreas (1987), while model 2 (dashed line) is the prediction of Smeets and van den Broeke (2008). Mean log value of $z_0 v$ specific to each season (Table 1) is used to calculate $Re_*$.

.

The findings for $Q_H$ from the 2012 season, which has more data points, agree well with the 2010 season: $C_{log}\ u_*$, and $C_{SR}\ u_*$ methods produced the lowest RMSE and the highest correlation coefficients among all the K-approach methods with observed variables (Table 3). These results suggest that in order to better simulate $Q_H$ with the bulk method, getting correct estimates of $u_*$ is more important than getting correct estimates of $\frac{z_v}{L}$. The variance error (scatter of residuals around 1:1

line) in the $C_{log}\ u_*$ method is smaller than in the $C_{M-O}\ \frac{z}{L}$ method (Fig. 7). The OPEC-derived $u_*$ underestimates $Q_H$ over the entire period (MBE < 0). Relative to the common K-approach with modeled $u_*$ and modeled $\frac{z_v}{L}$, the K-approach with the observed variables yields smaller RMSE, especially for $C_{log}$ method where the RMSE is more than halved. The $C_{log}\ u_*$ method outperforms any parameterization with the M-O stability corrections, even in the case when both $u_*$ and $\frac{z}{L}$ are measured ($C_{M-0}\ u_*\ \frac{z}{L}$, $C_{SR}\ u_*\ \frac{z}{L}$). The results for $Q_E$, for both years accord with the results for $Q_H$: using measured instead

of modeled $u_*$ improves the performance of the bulk method. Despite being less affected by the M-O stability corrections than the modeled sensible heat fluxes are, the modeled latent heat fluxes more closely resemble the observed ones if the OPEC-derived rather than modeled $\frac{z_v}{L}$ are used. In summary, $C_{log}$ with measured $u_*$ outperforms the other parameterizations but underestimates the sensible heat flux (MBE = -13.4 W m$^{-2}$ for 2012 season, Table 3). In contrast, $C_{log}$ with modeled $u_*$ significantly overestimates the sensible heat fluxes (MBE = 23.2 W m$^{-2}$ for the 2012 season, Table 3).

**3.2.7  Evaluation results: K-approach with new M-O stability functions ($C_{M-O}\ new$)**

In the $C_{M-O}$ method with modeled or OPEC-derived (measured) $\frac{z_{v,t}}{L}$, we assumed that the most commonly used M-O stability functions in glacier studies (Holtslag and de Bruin, 1988; Dyer, 1974) are applicable to our site. Our next step is to derive the new integrated stability functions, $\Psi_v(\frac{z_v}{L})$ and $\Psi_t(\frac{z_t}{L})$ to examine how well the optimization of the functions to specifically



**Table 3.** Results of the comparison between modeled and OPEC-derived sensible ($Q_H$) and latent ($Q_E$) heat fluxes, expressed as root-mean-square-error (RMSE), mean bias error (MBE; modeled minus observed), and Pearson correlation coefficient (r) for 2012 and 2010 (values in parenthesis) observational period, given for a range of methods (**1**) to (**23**) used to model the fluxes (see text).

| Method | $Q_H$ RMSE W m$^{-2}$ | $Q_H$ MBE W m$^{-2}$ | $Q_H$ r | $Q_E$ RMSE W m$^{-2}$ | $Q_E$ MBE W m$^{-2}$ | $Q_E$ r |
|---|---|---|---|---|---|---|
| (1) $C_{log}$ | 33.7 ( 30.6 ) | 23.2 ( 24.2 ) | 0.73 ( 0.53 ) | 15.7 ( 12.2 ) | 8.4 ( 10.9 ) | 0.90 ( 0.72 ) |
| (2) $C_{Rib}$ | 26.1 ( 22.7 ) | 7.6 ( 7.6 ) | 0.67 ( 0.43 ) | 12.9 ( 8.7 ) | 6.7 ( 6.1 ) | 0.87 ( 0.56 ) |
| (3) $C_{M-O}$ | 29.9 ( 26.7 ) | 17.1 ( 17.9 ) | 0.71 ( 0.50 ) | 14.4 ( 10.7 ) | 7.7 ( 9.1 ) | 0.89 ( 0.66 ) |
| (4) $C_{SR}$ | 25.9 ( 30.6 ) | 14.1 ( 23.6 ) | 0.73 ( 0.51 ) | 11.7 ( 12.8 ) | 6.4 ( 11.3 ) | 0.90 ( 0.69 ) |
| (5) $C_{log}\ u_*$ | 19.3 ( 11.6 ) | -13.4 ( -5.8 ) | 0.85 ( 0.81 ) | 5.6 ( 2.8 ) | 2.0 ( 0.6 ) | 0.95 ( 0.86 ) |
| (6) $C_{Rib}\ u_*$ | 27.3 ( 19.8 ) | -22.8 ( -16.1 ) | 0.82 ( 0.74 ) | 6.8 ( 4.1 ) | 1.4 ( -2.2 ) | 0.94 ( 0.77 ) |
| (7) $C_{M-O}\ u_*$ | 22.2 ( 13.5 ) | -16.7 ( -8.9 ) | 0.83 ( 0.80 ) | 5.7 ( 2.9 ) | 1.6 ( -0.4 ) | 0.95 ( 0.85 ) |
| (8) $C_{SR}\ u_*$ | 20.2 ( 10.2 ) | -14.6 ( -2.1 ) | 0.85 ( 0.81 ) | 5.8 ( 3.5 ) | 1.6 ( 1.9 ) | 0.95 ( 0.86 ) |
| (9) $C_{M-O}\ \frac{z}{L}$ | 21.3 ( 18.3 ) | -2.8 ( 3.9 ) | 0.71 ( 0.50 ) | 9.2 ( 6.5 ) | 3.9 ( 3.9 ) | 0.90 ( 0.63 ) |
| (10) $C_{SR}\ \frac{z}{L}$ | 19.9 ( 19.6 ) | -4.9 ( 8.6 ) | 0.73 ( 0.52 ) | 7.9 ( 7.7 ) | 3.1 ( 5.5 ) | 0.91 ( 0.66 ) |
| (11) $C_{M-O}\ u_*\ \frac{z}{L}$ | 25.1 ( 15.5 ) | -19.2 ( -10.6 ) | 0.79 ( 0.75 ) | 6.0 ( 3.3 ) | 1.1 ( -1.0 ) | 0.94 ( 0.82 ) |
| (12) $C_{SR}\ u_*\ \frac{z}{L}$ | 23.3 ( 12.1 ) | -17.3 ( -3.8 ) | 0.80 ( 0.74 ) | 6.1 ( 3.4 ) | 1.1 ( 1.2 ) | 0.95 ( 0.82 ) |
| (13) $C_{M-O}\ new\ \frac{z}{L}$ | 18.9 ( 16.7 ) | 0.3 ( 4.3 ) | 0.75 ( 0.54 ) | 7.9 ( 4.8 ) | 3.0 ( 1.0 ) | 0.90 ( 0.62 ) |
| (14) $C_{M-O}\ new$ | 27.8 ( 24.2 ) | 10.2 ( 10.3 ) | 0.66 ( 0.43 ) | 13.1 ( 9.0 ) | 6.2 ( 4.9 ) | 0.84 ( 0.45 ) |
| (15) $C_{kat}$ | 19.5 ( 20.8 ) | -4.6 ( 10.6 ) | 0.76 ( 0.50 ) | 7.9 ( 4.6 ) | 3.1 ( 2.8 ) | 0.92 ( 0.76 ) |
| (16) $K_{Int}\ K_{max}\ \frac{z}{L}$ | 29.1 ( 18.1 ) | -15.5 ( -2.8 ) | 0.55 ( 0.46 ) | 7.2 ( 4.2 ) | 0.1 ( 0.3 ) | 0.91 ( 0.69 ) |
| (17) $K_{Int}\ H_K\ \frac{z}{L}$ | 29.0 ( 18.1 ) | -15.4 ( -2.8 ) | 0.55 ( 0.46 ) | 7.2 ( 4.2 ) | 0.1 ( 0.3 ) | 0.91 ( 0.69 ) |
| (18) $C_{log}\ K_{max}\ \frac{z}{L}$ | 22.9 ( 15.2 ) | -15.1 ( -6.0 ) | 0.75 ( 0.59 ) | 6.8 ( 3.7 ) | 2.1 ( 0.6 ) | 0.93 ( 0.74 ) |
| (19) $C_{log}\ H_K\ \frac{z}{L}$ | 22.9 ( 15.2 ) | -15.0 ( -6.0 ) | 0.75 ( 0.59 ) | 6.8 ( 3.7 ) | 2.1 ( 0.6 ) | 0.93 ( 0.74 ) |
| (20) $C_{M-O}\ new\ K_{max}\ \frac{z}{L}$ | 15.6 ( 13.6 ) | -2.0 ( 4.1 ) | 0.81 ( 0.65 ) | 6.8 ( 3.7 ) | 2.1 ( 0.6 ) | 0.93 ( 0.74 ) |
| (21) $C_{M-O}\ new\ H_K\ \frac{z}{L}$ | 15.6 ( 13.6 ) | -1.9 ( 4.1 ) | 0.81 ( 0.65 ) | 6.8 ( 3.7 ) | 2.1 ( 0.6 ) | 0.93 ( 0.74 ) |
| (22) $C_{log}\ K_{max}$ | 23.6 ( 21.4 ) | 5.8 ( 8.9 ) | 0.59 ( 0.36 ) | 11.5 ( 8.2 ) | 5.2 ( 6.1 ) | 0.88 ( 0.61 ) |
| (23) $C_{log}\ H_K$ | 23.8 ( 21.5 ) | 6.2 ( 9.0 ) | 0.58 ( 0.36 ) | 11.7 ( 8.2 ) | 5.2 ( 6.1 ) | 0.87 ( 0.61 ) |

fit our data can improve the performance of the $C_{M-O}$ method. To do so, we first apply Eq. (5) to derive $\Psi_v$ using measured $u_*$, mean log $z_{0v}$ (Table 1), and measured $U_{z_v}$. We fit a second-order polynomial (Fig. 8) which represents the new integrated stability function for momentum flux, $\Psi_v(\frac{z_v}{L})$. Similarly, we derive $\Psi_t$ via Eq. (6) using measured $\Theta_*$, mean log $z_{0t}$ (Table 1), and measured $T_{z_t}$. The best fit polynomial in the plot, showing a relation between derived $\Psi_t$ and $\frac{z_t}{L}$ (Fig. 8), represents
5  the new integrated stability function for the sensible heat flux, $\Psi_t(\frac{z_t}{L})$. In this fitting exercise we used an additional condition that $\Psi_{v,t}(\frac{z_{v,t}}{L} = 0) = 0$. The results yield the following empirical expressions for the new stability corrections applicable to

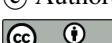


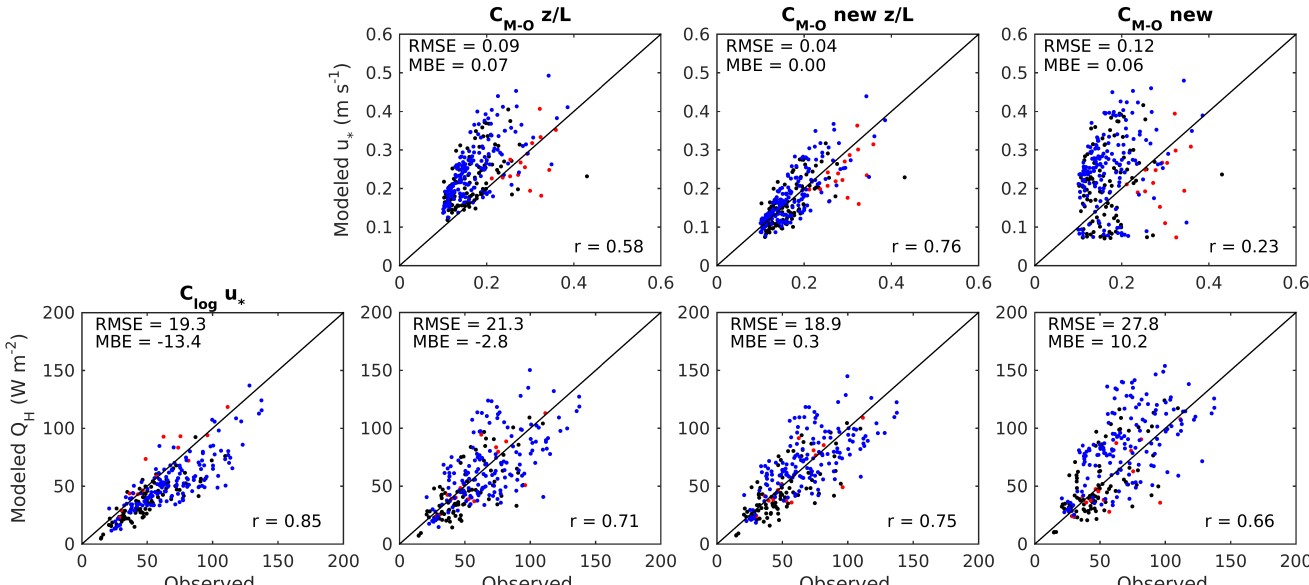

**Figure 7.** Comparison of 30-min values of observed (OPEC derived) and modeled (bulk method) shear stress (friction velocity, $u_*$) and sensible heat flux ($Q_H$) using the parameterization schemes in the bulk method (see text): $C_{log}\, u_*$, $C_{M-O}\, \frac{z}{L}$, $C_{M-O}\, new\, \frac{z}{L}$, and $C_{M-O}\, new$. Red points show values during the neutral conditions ($|\frac{z_{v,t}}{L}| < 0.1$) only, and blue points indicate values during the prevailing katabatic flow. Root-mean-square-error (RMSE; W m$^{-2}$), mean bias error (MBE; W m$^{-2}$), and Pearson correlation coefficient (r) are provided for each scatter plot.

$0 < \frac{z_{v,t}}{L} < 1$ conditions:

$$\Psi_v\left(\frac{z_v}{L}\right) = 7.79\left(\frac{z_v}{L}\right)^2 - 18.3\left(\frac{z_v}{L}\right), \tag{25}$$

$$\Psi_t\left(\frac{z_t}{L}\right) = -4.18\left(\frac{z_t}{L}\right)^2 - 8.68\left(\frac{z_t}{L}\right). \tag{26}$$

Due to few data points available during the very stable conditions ($\frac{z_{v,t}}{L} > 1$), the z-less scaling is assumed following the common approaches for a very stable atmospheric boundary layer (Foken, 2008). Therefore the new stability functions give: $\Psi_v(\frac{z_v}{L} > 1) = \Psi_v(\frac{z_v}{L} = 1) = -10.51$, and $\Psi_t(\frac{z_t}{L} > 1) = \Psi_t(\frac{z_t}{L} = 1) = 4.50$. The goodness-of-fit is calculated by regressing the modeled versus the OPEC-derived $\Psi_v$ and $\Psi_t$ values, giving statistically significant correlations (p-values < 0.01) with r=0.48 and r=0.35, respectively. The fitted polynomials, i.e. the coefficients in Eq. (25) and (26) are derived from the 2012 data only and then validated with 2010 data. The statistically significant correlations (p-values < 0.01) between modeled and observed values for $\Psi_v(\frac{z_v}{L})$ and $\Psi_t(\frac{z_t}{L})$ for the validation data (r=0.43 and r=0.41, respectively) demonstrate that the new stability corrections are successfully applicable to both seasons. Errors due to self-correlation (e.g., Klipp and Mahrt, 2004) are probably present in our dataset, however, we assume them negligible since we eliminated the non-stationary turbulence during which these errors can be large. Due to insufficiently large data sample for measured $Q_E$, we do not derive a new stability function



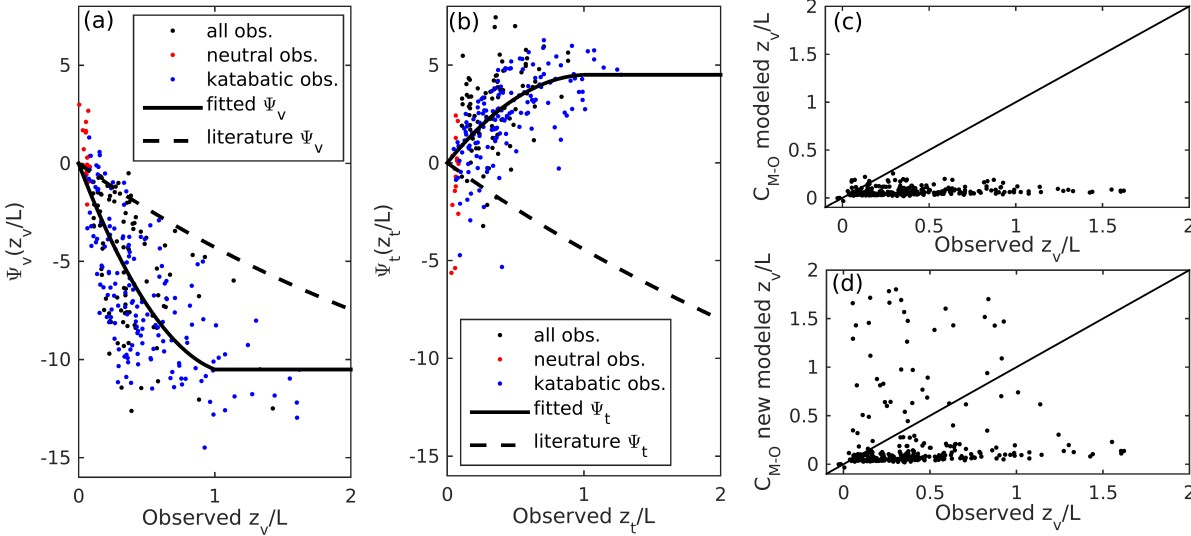

**Figure 8.** Panels (a) and (b): OPEC-derived 30-min values for stability corrections $\Psi_v$ and $\Psi_v$ versus OPEC-derived $\frac{z_{v,t}}{L}$, and the best fit polynomials representing the new integrated stability functions $\Psi_v(\frac{z_v}{L})$ and $\Psi_t(\frac{z_t}{L})$ (solid line), in comparison to the stability functions from the literature (dashed line). Red points show values during the neutral conditions ($|\frac{z_{v,t}}{L}| < 0.1$) only, and blue points indicate values during the prevailing katabatic flow. In (c): 30-min modeled $\frac{z_{v,t}}{L}$, via the fixed-point iteration method applied in $C_{M-O}$, versus 30-min OPEC-derived $\frac{z_{v,t}}{L}$. In (d): same as in (c) but for $C_{M-O}\ new$, which uses the new integrated stability functions. All data are from 2012 season.

for humidity ($\Psi_q(\frac{z_t}{L})$), nor assume that it is equal to the newly derived $\Psi_t$. Instead, we set $\Psi_q(\frac{z_t}{L})$ to zero since the stability seems to play a less important role in the $Q_E$ flux modification relative to $Q_H$ flux modification, as shown for the K-approach with observed variables (Section 3.2.6).

5    Finally, we test the performance of the $C_{M-O}$ method with the new stability functions, yielding: (**13**) $C_{M-O}\ new\ \frac{z}{L}$ method that considers the OPEC-derived $\frac{z_{v,t}}{L}$, and (**14**) $C_{M-O}\ new$ method that takes modeled $\frac{z_{v,t}}{L}$ via the fixed-point iteration method of Munro (1989). As expected, the new stability corrections provide a better match between modeled and observed sensible heat fluxes than in the case with the stability corrections from literature (Fig. 7). If the modeled $\frac{z_{v,t}}{L}$ is used, however, the improvement of the bulk method performance is almost negligible. Despite the fact that the fixed-point iteration method

10  successfully converges to a solution, the predicted solutions for $\frac{z_{v,t}}{L}$ poorly resemble the OPEC-derived values. The modeled $\frac{z_v}{L}$ applied in the $C_{M-O}$ and $C_{M-O}\ new$ methods are significantly lower than the OPEC-derived $\frac{z_v}{L}$ (Fig.7). Because the static stability $\frac{z_{v,t}}{L}$ is consistently underestimated in the bulk methods with the fixed-point iterative scheme, the stability corrections have a very small effect in modifying the modeled turbulent heat fluxes.



### 3.3 Methods based on katabatic flow models

Here we describe in more details our two types of bulk methods that are rooted in the katabatic flow models, and present the evaluation results:

#### 3.3.1 Katabatic $C_{kat}$ method

Rather than integrating the flux-gradient relations with a chosen parametrization for $K$, Oerlemans and Grisogono (2002) derived a bulk approach for surface heat fluxes through a simplified scaling of the governing equations for heat and momentum balance in a 1-D katabatic flow model (Prandtl, 1942). Their basic assumption was that the katabatic flow is characterized by a well defined wind maximum, setting the exchange coefficient for heat proportional to the maximum wind speed and to the height of the wind maximum. With this scaling approach they determined a 'katabatic bulk exchange parameter' for heat flux

($C_{kat}$, in m s$^{-1}$) as:

$$C_{kat} = -k \, k_2^2 \, \Theta_s \left( \frac{g}{T_0 \, \gamma \, Pr} \right)^{1/2}, \tag{27}$$

where $k$ and $k_2$ are dimensionless empirical constants, $\Theta_s$ is the temperature deficit at the glacier surface (negative value; $\theta(0) = \Theta_s$, $\theta(z \to \infty) = 0$), $\gamma$ is the background potential temperature lapse rate, and $Pr$ denotes the eddy Prandtl number $\left( \frac{K_M}{K_H} \right)$. Defining the exchange coefficient in this way makes the sensible heat flux to increase quadratically with the surface

temperature deficit:

$$Q_H = \rho_a \, c_p \, C_{kat} \, \Theta_s. \tag{28}$$

$\Theta_s$ can be replaced by 2 m air-surface temperature difference ($\Delta T$), but it gives only a crude estimate of the actual temperature deficit. The latent heat flux is similarly expressed as:

$$Q_E = \frac{0.622}{p} \, \rho_a \, L_v \, C_{kat} \, (e_{z_t} - e_0). \tag{29}$$

Neither $Q_H$ nor $Q_E$ explicitly depend on $u_*$ because the momentum flux and wind speed are obtained from the katabatic flow model in which the eddy viscosity is parametrized through the background variables ($\Theta_s$ and $\gamma$). $C_{kat}$ becomes smaller as the atmospheric boundary layer becomes more stable (larger $\gamma$) because the katabatic flow weakens with increasing $\gamma$. As noted in Oerlemans and Grisogono (2002), this is a dynamic effect which should not be confused with the effect of stratification on the exchange coefficients as predicted by the M-O theory.

#### 3.3.2 Evaluation results: $C_{kat}$ method

Our data are not suitable to adequately test this bulk method since we do not have the vertical profile observations of potential temperature needed to estimate $\Theta_s$ and $\gamma$. Nevertheless, our data are sufficient to investigate the validity of the quadratic relation between $Q_H$ and $\Delta T$. In particular, assuming that the quadratic fit holds, we will optimize $C_{kat}$ so that the modeled $Q_H$ gives the best fit to the observed $Q_H$. Although this optimization can be performed in multiple different ways, we chose





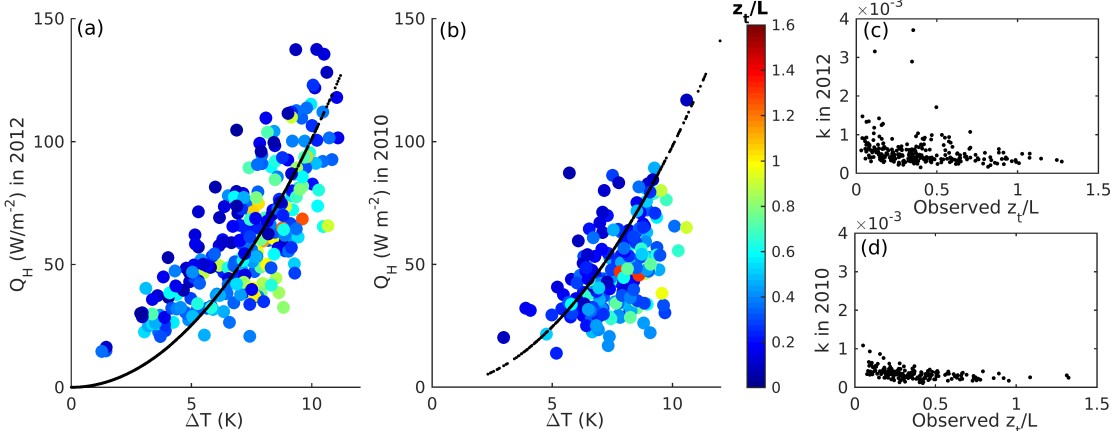

**Figure 9.** Observed 30-min sensible heat fluxes ($Q_H$) versus observed air-surface temperature difference ($\Delta T$), and modeled 30-min $Q_H$ fluxes (black dotted line) according to the $C_{kat}$ method, for (a) 2012 and (b) 2010 season. Each observation point is colored according to its 30-min value of $\frac{z_t}{L}$ (colorbar). Panels (c) and (d) show the parameter $k$, derived from 30-min observations of $Q_H$ and $\Delta T$, versus OPEC-derived $\frac{z_t}{L}$ for the 2012 and 2010 seasons, respectively.

to set all the parameters constant except one used for model tuning. The values of constant parameters are taken from the field data from a 10 km long valley glacier in the Austrian Alps (PASTEX-94; Greuell and Struijk, 1994): $k_2=1$, $\gamma=0.005$ K m$^{-1}$, $Pr=2$ and $T_0=273$ K. The chosen optimization parameter $k$ is then evaluated by minimizing the RMSE between modeled and observed 30-min $Q_H$ for each season separately, using the air-surface temperature difference ($T_{z_t} - T_0$) as a substitute for $\Theta_s$.

The optimization of the $k$ parameter in this method, as our (**15**) $C_{kat}$ method, yields $k = 4.12 \times 10^{-4}$ and $k = 4.00 \times 10^{-4}$ for 2012 and 2012 season, respectively. When this optimized $C_{kat}$ is used to derive sensible and latent heat fluxes (Eq. (28) and (29)) the modeled fluxes simulate the observed ones better (smaller RMSE and MBE) than any of the K-approach methods with common parameterizations (**1**)-(**4**) (Table 3). There is a quadratic relation between OPEC-derived $Q_H$ and $\Delta T$ (Fig. 9),

10 but this relation only explains 58 % of the variance in the observations. Our results do not guarantee the validity of Eq. (28), but demonstrate that the simple katabatic model of Oerlemans and Grisogono (2002) reproduces the basic characteristics of katabatic flow at our site; this model can simulate the turbulent heat exchange more successfully than the K-approach method. Keeping all the parameters, except $k$, constant in Eq. (27) and using the observed 30-min values for $Q_H$ and $\Delta T$ in Eq. (28), we find that $k$ (and therefore $C_{kat}$) decreases as the static stability ($\frac{z_{v,t}}{L}$) increases, however, the regression between $k$ and

15 $\frac{z_{v,t}}{L}$ is not statistically significant. The decrease in $C_{kat}$ as the atmosphere becomes more stable is expected according to the katabatic flow model (Oerlemans and Grisogono, 2002).



### 3.3.3 Katabatic $K_{Int}$ method ($K_{Int}$ $K_{max}$, $K_{Int}$ $H_K$)

Equivalently to the $C_{kat}$ method, the second method assumes that the katabatic flow dominates at the site throughout the observational period. Following Grisogono and Oerlemans (2001) and Parmhed et al. (2004) we use the assumed linear-Gaussian profile for eddy viscosity $K(z)$ as

$$K(z) = \frac{K_{max}\ \exp(0.5)}{H_K}\ z\ \exp\left[-0.5\left(\frac{z}{H_K}\right)^2\right],\tag{30}$$

where $K_{max}$ is the maximum value of $K(z)$ (in m$^2$ s$^{-1}$) reached at the height $H_K$ (in m) above the surface. Near the surface, this parameterization for $K(z)$ is similar to O'Brien's cubic polynomial approximation applicable to neutral and stable boundary layer (O'Brien, 1970; Stull, 1988). In the original model of Grisogono and Oerlemans (2001), $K_{max}$ and $H_K$ are related to the background variables ($\gamma$, $\Theta_s$), slope of the surface, and $Pr$. To adequately determine $K_{max}$ and $H_K$, detailed observations of the wind speed and temperature profiles with height are required. Setting $Pr$=1 (thus setting $K_M$=$K_H$=$K_E$), inserting the parameterizations for $K(z)$ into the bulk-gradient relations (Eq. (10) - (11)) and integrating them with respect to $z$ (treating $K_{max}$ and $H_K$ as constants), we obtain the following bulk expressions for the turbulent fluxes:

$$u_* = \left(\frac{U_{z_v}}{K_{Int}}\right)^{1/2},\tag{31}$$

$$Q_H = \rho_a\ c_p\ \frac{T_{z_t} - T_0}{K_{Int}},\tag{32}$$

$$Q_E = \frac{0.622}{p}\ \rho_a\ L_v\ \frac{e_{z_t} - e_0}{K_{Int}},\tag{33}$$

where

$$K_{Int} = \int_{z_{0v}}^{z_v} \frac{\mathrm{d}z}{K(z)}\ \approx\ \frac{H_K}{2\ K_{max}\ \exp(0.5)}\left[2\ln\left(\frac{z_v}{z_{0v}}\right) + \frac{0.5}{H_K^2}\left(z_v^2 - z_{0v}^2\right)\right],\tag{34}$$

which is valid for $z_v < H_K$. In the absence of vertical profile observations we make use of the OPEC data (i.e. 30-min values for $u_*$), which when inserted in Eq. (31) together with observed $U_{z_v}$ can give the best estimate for measured $K_{Int}$ (in s m$^{-1}$). Using the measured $K_{Int}$ in Eq. (34), and the log mean value (for neutral conditions only) of $z_{0v}$ we derive: (a) measured values for $K_{max}$ when $H_K$ is set to a constant value of $\overline{H_k}$=20 m; and (b) measured values for $H_K$ when $\overline{K_{max}}$ = 0.8 m$^2$ s$^{-1}$. The assumed constant values for $\overline{K_{max}}$ and $\overline{H_K}$ are taken from their frequently observed values from a field study at a glacier in Iceland (Parmhed et al., 2004) as this was the only study we found that directly measured the two parameters. We also investigate whether the measured $K_{max}$ and $H_K$ can be expressed as functions of the OPEC-derived $\frac{z_v}{L}$. If there is a strong empirical relation, this will indicate that from the local stability metric one can constrain the parameters in the katabatic model.





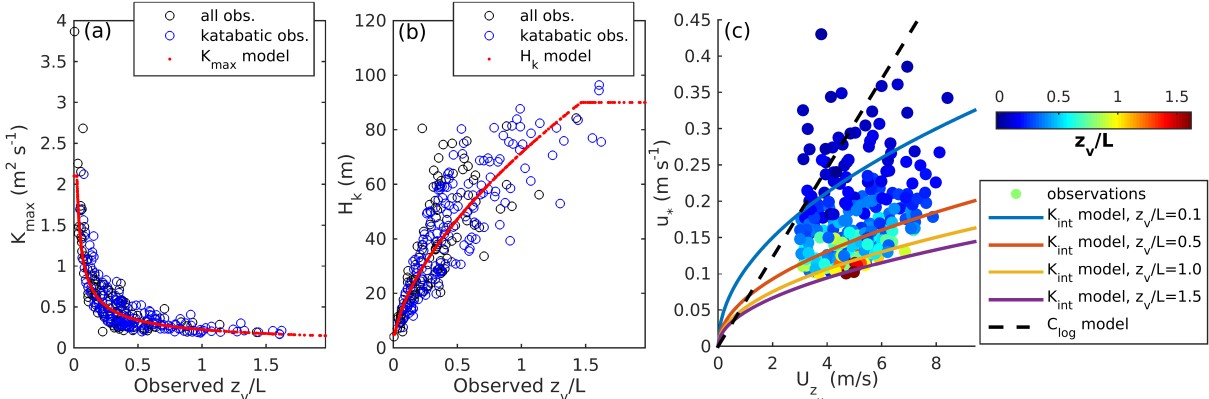

**Figure 10.** $K_{max}$ (panel a) and $H_k$ (panel b), calculated from 30-min OPEC-derived $u_*$ and observed $U_{z_v}$, plotted against OPEC-derived $\frac{z_v}{L}$ for 2012 season. Black circles represent all observations, blues circles are observed values during prevailing katabatic conditions, and red dotted line is the empirical function for $K_{max}(\frac{z_v}{L})$ (panel a) and $H_K(\frac{z_v}{L})$ (panel b). Panel (c): OPEC-derived 30-min friction velocity, $u_*$, versus observed wind speed $U_{z_v}$, for 2012 season, where each observation point is colored according to its 30-min value of $\frac{z_v}{L}$ (colorbar). Solid lines are modeled $u_*$, using $K_{Int}$ parameterization with $\overline{H_k}$=20 m and modeled $K_{max}$, for four different stability conditions ($\frac{z_v}{L}$=0.1, 0.5, 1.0, 1.5). Dashed line is modeled $u_*$ using the bulk method with $C_{log}$ parameterization.

### 3.3.4 Evaluation results: $K_{Int}$ method ($K_{Int}\ K_{max}$, $K_{Int}\ H_K$)

The relation between the measured 30-min $K_{max}$ ($H_K$) and the OPEC-derived $\frac{z_v}{L}$ is strongly non linear (Fig. 10). The best fitting models (Fig. 10) have the following power-law relations:

$$K_{max}\Big(\frac{z_v}{L}\Big) = K_{max0}\Big(\frac{z_v}{L}\Big)^{-0.60}, \tag{35}$$

$$H_K\Big(\frac{z_v}{L}\Big) = H_{K0}\Big(\frac{z_v}{L}\Big)^{0.60}, \tag{36}$$

where $K_{max0}$=0.22 m$^2$ s$^{-1}$, and $H_{K0}$=71.52 m. Note that these empirical fits are assumed valid within the range of observed values for $K_{max}$ and $H_K$ taken from Parmhed et al. (2004): 0.03 m$^2$ s$^{-1}$ < $K_{max}$ < 2.1 m$^2$ s$^{-1}$, and 5 m < $H_K$ < 90 m. The power-law models are optimized on the 2012 data only (Fig. 10), while the 2010 data is used for the model validation. The validation yields high correlation coefficients between the 30-min modeled and observed values: r=0.89 for the $K_{max}$ values, and r=0.83 for the $H_K$ values. This result indicates that, for our study site, the same empirical model is equally applicable to both field seasons.

The static stability drives the strength of the proportionality between measured friction velocity and wind speed (Fig. 10), so that the higher the stability, the smaller the constant of proportionality between $u_*$ and $U_{z_v}$. The modeled $u_*$ derived from Eq. (31), where $K_{Int}$ is evaluated using $\overline{H_k}$=20 m and the $K_{max}$ empirical model (Eq. (35)), successfully represents two observational features: (i) a non-linear relation between friction velocity and wind speed; and (ii) a functional dependence of



this non-linear relation on $\frac{z_v}{L}$. For comparison, we also show modeled $u_*$ versus $U_{z_v}$ where the $C_{log}$ method (K-approach) is used to determine the friction velocity (Eq. (14)). As expected, for the near-neutral conditions ($|\frac{z_v}{L}| < 0.1$), the K-approach works relatively well, because of the near-linear dependence between the OPEC-derived $u_*$ and wind speed. However, as soon as the stability increases, the linearity is replaced by a relation that more closely resembles $u_* \propto \sqrt{U_{z_v}}$ (i.e. momentum flux proportional to wind speed) where the strength of the proportionality depends on the static stability. The same results are found for 2010 season. Finally, we compare the modeled versus observed turbulent fluxes incorporating: (**16**) $K_{Int}\ K_{max}\ \frac{z}{L}$ method, which takes the empirical relation for $K_{max}$ (Eq. (35)) and the OPEC-derived $\frac{z_v}{L}$, and (**17**) $K_{Int}\ H_K\ \frac{z}{L}$ method, which takes the empirical relation for $H_K$ (Eq. (36)) and the OPEC-derived $\frac{z_v}{L}$. The results show that both methods have almost the same performance, and they give the best estimate of the friction velocity or momentum flux (Fig. 11) among all the bulk schemes we tested so far. The $K_{Int}$-approach, however, does not perform as well for the sensible and latent heat fluxes as it does for the friction velocity: correlation between observed and modeled fluxes are the lowest among the methods tested (Table 3). We conclude that, while the linear-Gaussian parametrization of $K(z)$ works well for the momentum flux, it works poorly for the heat flux, indicating a shortcoming in the commonly used assumption that $K_M$ and $K_H$ can share the same parametrization.

### 3.4 Hybrid methods with $K_{Int}$ and K-approach ($C_{log}\ K_{max}, C_{log}\ H_K, C_{M-O}\ K_{max}, C_{M-O}\ H_K$)

Our final group of bulk methods reflect a 'hybrid' approach where the friction velocity is assessed from the $K_{Int}$ method (Eq. (34)), while the heat fluxes are calculated from the K-approach $C_{log}$ method (or $C_{M-O}\ new$ method) following Eq. (18) and (19):

$$Q_H = \rho_a\ c_p\ C_{t,log}\ u_*\ (T_{z_t} - T_0), \tag{37}$$

$$Q_E = \frac{0.622}{p}\ \rho_a\ L_v\ C_{q,log}\ u_*\ (e_{z_t} - e_0). \tag{38}$$

In this way, we assume that the linear-Gaussian profile of $K(z)$ dictates the solution for the wind speed profile with height (Grisogono and Oerlemans, 2001), while the sensible and latent turbulent heat fluxes are calculated assuming the validity of the logarithmic vertical profiles for temperature and specific humidity (Prandtl, 1942). The performance of the following six bulk schemes are tested, where '$\frac{z}{L}$' notation indicates that OPEC-derived static stability is used : (**18**) $C_{log}\ K_{max}\ \frac{z}{L}$ method, with the empirical function for $K_{max}(\frac{z_v}{L})$, (**19**) $C_{log}\ H_k\ \frac{z}{L}$ method, with the empirical function for $H_k(\frac{z_v}{L})$, (**20**) $C_{M-O}\ new\ K_{max}\ \frac{z}{L}$ method and (**21**) $C_{M-O}\ new\ H_k\ \frac{z}{L}$ method with the new stability function $\Psi_t(\frac{z_t}{L})$ in the parameterization for $C_{M-O,t}$. The final two are (**22**) $C_{log}\ K_{max}$ method and (**23**) $C_{log}\ H_K$ method, which share the same formulation as (**18**) and (**19**), respectively, but use the fixed-point iterative method of Munro (1989) to derive $\frac{z_v}{L}$.

### 3.4.1 Evaluation results: hybrid methods ($C_{log}\ K_{max}, C_{log}\ H_K, C_{M-O}\ K_{max}, C_{M-O}\ H_K$)

The hybrid methods outperform the $K_{Int}$ methods in simulating the $Q_H$ fluxes, and outperform the K-approach with the common parameterizations in simulating all the fluxes (Fig. 11). The best performance for $Q_H$ fluxes arises from the $C_{M-O}\ new\ H_k\ \frac{z}{L}$ method, but the application of the new stability corrections only make sense if the OPEC-derived $\frac{z_{v,t}}{L}$ is used. Without a priori knowledge of $\frac{z_{v,t}}{L}$, but using the proposed fixed-point iterative method to derive $L$, the skill in simulating $u_*$ and $Q_H$ with the



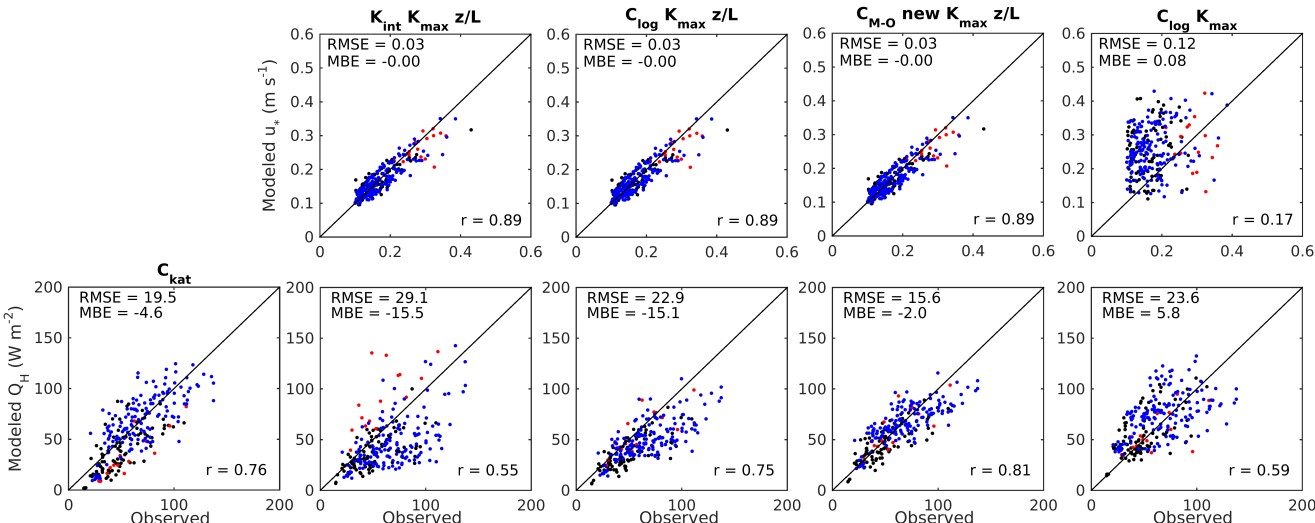

**Figure 11.** Comparison of 30-min values of observed (OPEC derived) and modeled friction velocity ($u_*$) and sensible heat flux ($Q_H$) using different bulk approaches (see text): $C_{kat}$, $K_{Int}\ K_{max}\ \frac{z}{L}$, $C_{log}\ K_{max}\ \frac{z}{L}$, $C_{M-O}\ new\ K_{max}\ \frac{z}{L}$, and $C_{log}\ K_{max}$. Red points show values during the neutral conditions ($|\frac{z_{v,t}}{L}| < 0.1$) only, and blue points indicate values during the prevailing katabatic flow. Root-mean-square-error (RMSE; W m$^{-2}$), mean bias error (MBE; W m$^{-2}$), and Pearson correlation coefficient (r) are provided for each scatter plot.

hybrid methods drops. This degradation of skill is shown by $C_{log}\ K_{max}$ and $C_{log}\ H_K$ methods, both of which yield higher RMSE than their equivalents $C_{log}\ K_{max}\ \frac{z}{L}$ and $C_{log}\ H_K\ \frac{z}{L}$. The choice of $K_{max}(\frac{z_v}{L})$ or $H_K(\frac{z_v}{L})$ in the hybrid methods makes no different in the evaluation results (Table 3).

## 4 Discussion and Uncertainty Analysis

We summarize the inter-comparison results, i.e. error estimates, between modeled and observed fluxes, for a selection of the most relevant bulk approaches used in the study (Fig. 12). The error estimates are shown for the 2012 season only, and agree well with the estimates for the 2010 season. By the most relevant bulk approaches we consider those that use OPEC-derived roughness lengths and mean meteorological variables (e.g. 2-m temperature, wind speed, specific humidity averaged over 30-min time intervals) and modeled or OPEC-derived M-O stability parameter ($\frac{z_{v,t}}{L}$). Note that the schemes with OPEC-derived

$\frac{z_{v,t}}{L}$ do not strictly qualify as the bulk approaches, however, we include them in the analysis in order to distinguish between the actual and the empirical bulk model performance, where the empirical model uses OPEC-derived rather then modeled $\frac{z_{v,t}}{L}$. In implementing the bulk schemes, we assumed that the glacier surface has constant roughness lengths ($z_{0v,0t,0q}$), equal to the mean log values from filtered OPEC data for neutral conditions only (Table 1), throughout the observational period. The error estimates for each bulk scheme (Fig. 12) are presented as a Mean Square Error (MSE, which is equal to RMSE$^2$

from Table 3) decomposed into a squared MBE and a variance error (VE), satisfying MSE = MBE$^2$ + VE. We also show a



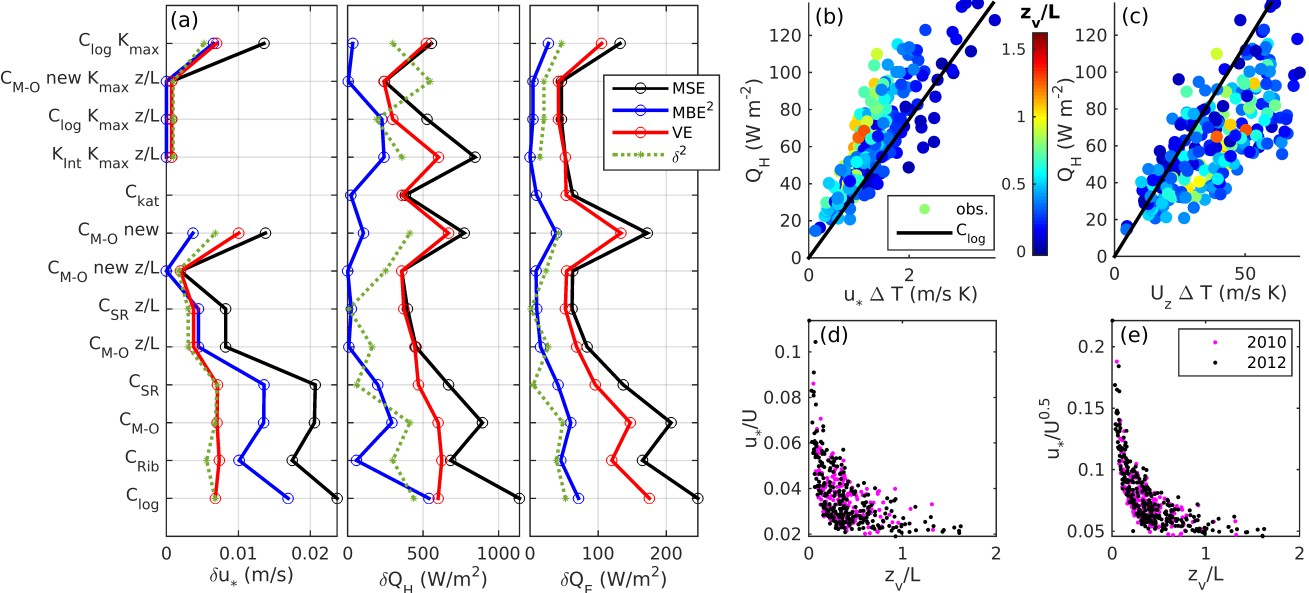

**Figure 12.** Panel (a): Mean Square Error (MSE), squared Mean Bias Error (MBE$^2$), Variance Error (VE) and squared model RMSE ($\delta^2$; derived from the Monte Carlo runs with perturbed roughness lengths - see text) for a selection of most relevant bulk approaches in the study, for 2012 season. Note that MSE = MBE$^2$ + VE. Panels (b) and (c): OPEC-derived 30-min sensible heat flux ($Q_H$) versus measured products $u_* \Delta T$ and $U_{z_v} \Delta T$, respectively, for 2012 season. Each point is colored according to its 30-min OPEC-derived $\frac{z_v}{L}$. Panels (d) and (e): Measured 30-min values of $\frac{u_*}{U_{z_v}}$ and $\frac{u_*}{\sqrt{U_{z_v}}}$, respectively, versus OPEC-derived $\frac{z_v}{L}$ for the two years.

squared model RMSE calculated from the individual 30-min error estimates (error bars in Fig. 5) over the whole observational period, reflecting the sensitivity of the bulk scheme to the uncertainty in the roughness lengths. The hybrid approach with the $K_{Int}$ method for assessing $u_*$ outperforms any other actual bulk model in the study (Fig. 12). The OPEC-derived $\frac{z_{v,t}}{L}$ significantly improves the simulations of the turbulent fluxes, in some cases reducing the MSE by more than 50 %. Again, the

5    empirical $K_{Int}$-approach performs the best in simulating the shear stress among all the empirical bulk approaches, while the empirical hybrid approach ($K_{Int}$ in combination with $C_{M-O}$ new), with the new stability correction $\Psi_t(\frac{z_t}{L})$, performs the best in simulating $Q_H$. For modeled $u_*$, the total error is dominated by the contribution from the mean bias error (MBS$^2$ > VE), while for the modeled $Q_H$ and $Q_E$ it is the variance error that dominates. The better performing bulk methods are those that are able to reduce MBE, while VE stays roughly unchanged across the methods. The VE gets reduced only when stability

10   corrections are acting on the OPEC-derived $\frac{z_{v,t}}{L}$, reveling that the remaining variability in the data, which is not explained by the variability in the mean meteorological variables (e.g. $U_z$, $\Delta T$), is likely explained by the variability in $\frac{z_{v,t}}{L}$. The assessment of $Q_H$ with the $C_{M-O}$ methods has the largest sensitivity to the perturbations in the roughness lengths, probably because the uncertainties in both $z_{0v}$ and $z_{0t}$ propagate into the modeled $\frac{z_{v,t}}{L}$ and then, via the fixed-point iterative method, back-propagate into the $Q_H$ estimates. The best overall performing model ($C_{M-O}$ new $K_{max}$ $\frac{z}{L}$) for $Q_H$ has the highest sensitivity to the





roughness length uncertainties. This is because the new stability functions are optimized using the constant values for $z_{0v,0t,0q}$ (Table 1), thus any deviation from these values leads to a suboptimal performance of the stability corrections and consequently a misrepresentation of $Q_H$.

In the remaining part of this section we discuss our results in the order of initially introduced sources of uncertainties in the bulk methods, particularly in simulating $Q_H$ on sub-daily scales. To reiterate, main sources of uncertainties originate from (1) assumed rather than measured surface temperature, (2) estimation of roughness lengths, (3) representation of stability corrections in the bulk methods, and (4) an absence of a successful model applicable during very stable conditions and prevailing katabatic flow on sloped glacier surfaces.

1. In the absence of reliable measurements of surface temperature, $T_0$, we assumed the glacier surface to be at the melting point throughout the observational period. We quantify the uncertainty of this assumption using the equivalent Monte Carlo approach we applied earlier to derive the model RMSE due to the roughness length uncertainties. For each Monte Carlo run, we randomly prescribe $T_0$ in each evaluation of 30-min $Q_H$ via the bulk method, assuming $T_0$ has a normal distribution with a mean of 0°C and a standard deviation of 0.5°C. In total, 1000 ensemble runs for each bulk method are produced, yielding an error estimate (a standard deviation from the ensemble) for each 30-min modeled $Q_H$. Model RMSE for $Q_H$ over the 2012 observational period, using the same filtered 30-min points as before, is in the range of 2 to 4 W m$^{-2}$ across all the bulk approaches. Despite being significant for the 30-min $Q_H$ estimates, the model error is negligible over the whole observational period, especially when compared with the RMSE between modeled and observed fluxes for each bulk method (Table 3). This error estimate does not account for any potential systematic biases in the surface temperature that might occur throughout the observational period (e.g. overnight cooling or refreezing of the surface; changes in surface temperature due to a debris cover or a formation of water channel at the study site). In other words, as long as the measured fluctuations of surface temperature are small, random, and centered at 0°C, using the assumed rather than measured values does not cause any substantial error in the simulated fluxes over the entire observational period. Furthermore, the meteorological measurements also carry random and systematic errors (e.g. radiative heating of naturally ventilated temperature sensor) which can propagate into the bulk method estimates of the turbulent fluxes. Our Monte Carlo uncertainty assessment, accounting for a small random errors in $T_0$, also represents the uncertainty due to small random errors in $T_z$, as long as there is no significant systematic bias in $T_z$. Finally, as this study facilitated only the on-glacier measurements, a question remains on how well the bulk approaches perform with the use of off-glacier measurements of mean meteorological variables.

2. To produce the accurate estimates of roughness lengths ($z_{0v}$, $z_{0t}$, $z_{0q}$) we used detailed eddy-covariance measurements subjected to a series of recommended corrections and filters. Since the K-approach with $C_{M-O}$ method is used to derive the roughness lengths, it is crucial to assess the roughness length during the conditions when the best performance of this method is expected (i.e. during the near-neutral stability conditions, $\frac{z_{v,t}}{L} < 0.1$). As anticipated, these conditions at our study site rarely occurred, resulting in a significantly reduced data sample. Working with the small data statistics, we assumed the mean log values of $z_{0v,0t,0q}$ as representative of the entire observational period. For both seasons, we

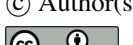



obtained $z_{0v}$ with the order of magnitude $10^{-3}$ m, which is characteristic for a hummocky ice, while our $z_{0t}$ and $z_{0q}$ estimates turned out to be two orders of magnitude smaller than $z_{0v}$. These findings are in agreements with the previous studies on glaciers (e.g., Smeets et al., 1998; Conway and Cullen, 2013), while the latter finding is corroborated with the surface renewal theory of Andreas (1987). Considering that we obtained roughly the same estimates for $z_{0v,0t}$ from the two different summers at the same site, gives some confidence in the robustness of the determination method. Taking into account the small data statistics, the absence of any model for predicting a time-varying roughness lengths, and the lack of any significant changes in the surface roughness during the observational period, we operated with the assumption of a constant $z_{0v,0t,0q}$ throughout each season. The uncertainty in the modeled turbulent fluxes due to potentially time-varying roughness lengths was quantified via the Monte Carlo approach (modeled RMSE in Fig. 12) and is found to be smaller than the RMSE between modeled and observed fluxes. Thus, the modeled fluxes are poorly simulated due to the poorly defined parameterizations, rather than the poorly constrained roughness lengths. In our Monte Carlo assessment of the model RMSE, we assumed the errors in 30-min $z_{0v,0t,0q}$ to be random and small, rather than having a systematic error due to any potential changes in the actual surface roughness (e.g. snow fall; development of a surface drainage system; inhomogeneous surface ablation). The assumed random errors are attributed to instrumental (measurement) error and to the choice of the threshold stability value for the near-neutral conditions (i.e. $\left|\frac{z_{v,t}}{L}\right| < 0.1$). As discussed below, we posit that the large variance error (the scatter in the graphs showing 30-min modeled versus observed turbulent fluxes) is mainly due to the variability in the meteorological mean variables (temperature, wind speed, local stability) rather than the variability in the roughness lengths in time. If, instead of the constant mean log values for $z_{0v,0t,0q}$, we use their observed values specific to each 30-min segment with the near-neutral conditions, the resulting overfit would not improve the performance over all the points because: (i) the overfitted points represent < 5 % of points in the observation period, and (ii) the model performs poorly outside the near-neutral conditions. Fitting the bulk method to the points with the near-neutral conditions is thus like overfitting a trendline model to few points that lie on a statistically insignificant trend of noisy data.

3. When testing the stability corrections in the bulk methods, we assumed that the main predictor of the time-varying bulk exchange coefficient is the time-varying local stability, not the changes in the surface roughness. Using the two common K-approach parameterizations to account for the stability corrections ($C_{Rib}$ and $C_{M-O}$ method) we found that neither scheme significantly improves modeling of turbulent fluxes relative to the scheme without the stability corrections ($C_{log}$ method). According to our analysis, there are two reasons for this: (i) the empirical stability functions ($\Psi_{v,t,q}$) taken from the literature perform poorly on our data, and (ii) if not directly measured via the eddy-covariance system, $\frac{z_{v,t}}{L}$ metric is impossible to accurately obtain because no functional relationship is found between the metric and the mean meteorological variables (e.g. temperature and wind speed). The latter reason also explains why measured $R_{ib}$ could not be expressed as a function of OPEC-derived $\frac{z_{v,t}}{L}$. The proposed stratification-dependent conversion of $\frac{z_{v,t}}{L}$ into $R_{ib}$ (e.g., Arya, 2001) thus does not work for our data. Applying our own empirically derived stability functions in the bulk method ($C_{M-O}\ new$) helped to somewhat resolve the issue (i), however, issue (ii) remains and can cause a substantial





overestimation of the turbulent fluxes (Fig. 7). More importantly, our results indicate that the stability corrections become important modifiers of $Q_H$ fluxes only if the correct predictors are used in the bulk model for $Q_H$. If $Q_H \propto U_{z_v} \Delta T$ is used (Fig. 12) for example, the variability of the observed $Q_H$ points around the modeled trendlines cannot be explained as a function of $\frac{z_{v,t}}{L}$. On the other hand, if $Q_H \propto u_* \Delta T$ and OPEC-derived $u_*$ is applied, the variability can be better

predicted by a function of $\frac{z_{v,t}}{L}$ (Fig. 12). In fact, this function is the new empirical stability correction $\Psi_t(\frac{z_t}{L})$ (Eq. (26)). Since the K-approach with common parameterizations for $C$ cannot successfully simulate $u_*$ (Fig. 5 and 7), it is not surprising that the K-approach, with or without stability corrections, fails to successfully simulate $Q_H$. Note that the new stability correction $\Psi_t(\frac{z_t}{L})$ acts in the opposite direction than those commonly used for glacier studies: in our case, the modeled $Q_H$ needs to increase, rather than be suppressed, as the stability increases. To summarize,

our findings corroborate previous experimental and theoretical evidence that the M-O theory is poorly applicable over sloping glacier surfaces (Denby and Greuell, 2000). However, in contrast to the findings in Conway and Cullen (2013) where the stability functions over-suppressed the turbulent heat fluxes in a stable atmosphere, we find that the stability functions under-suppress the fluxes. The tendency to under-suppress the fluxes with the $C_{M-O}$ method is also found for another glacier surface (Fitzpatrick et al., 2017). The M-O stability parameter ($\frac{z_{v,t}}{L}$), however, plays an important role

in modifying the turbulent heat fluxes if accurate estimates (e.g. OPEC data) of this parameter and $u_*$ are used in the bulk method. While modeling of $\frac{z_{v,t}}{L}$ remains a challenge, the $K_{Int}$ approach, further discussed below, can significantly improve the assessment of $u_*$.

4. As a general rule, under stable stratification and in the presence of low level jet (katabatic flow), intermittent turbulence and gravity waves are present, and steady-state conditions do not exist (e.g., Foken, 2008; Axelsen and van Dop, 2009).

In this study, in order to obtain reliable OPEC-derived fluxes, we removed the data which failed to satisfy the criteria of stationarity ('stationarity' filter). Assuming that our filtered data sample reflects the steady-state conditions, we found that the $K_{Int}$ approach can simulate the 30-min $u_*$ better than any K-approach if the OPEC-derived $\frac{z_{v,t}}{L}$ is used. The reason behind this successful performance of the $K_{Int}$ method is in the parametrization of $K(z)$: while the K-approach (where $K(z) \propto u_*$) relies on $u_* \propto U_{z_v}$, the $K_{Int}$ approach (where $K(z)$ is not dependent on $u_*$) gives $u_* \propto \sqrt{U_{z_v}}$.

A dependency of the 30-min observed $\frac{u_*}{\sqrt{U_{z_v}}}$ on the OPEC-derived $\frac{z_v}{L}$ (Fig. 12) is more accurately represented with a fitting function than a dependency of observed $\frac{u_*}{U_{z_v}}$ on $\frac{z_v}{L}$. Therefore, the empirical functions $K_{max}(\frac{z_v}{L})$ and $H_K(\frac{z_v}{L})$ (Eq. (35) and (36)) give a much better fit to the data than any stability function ($\Psi_v(\frac{z_v}{L})$) in the K-approach. Despite the absence of vertical profile measurements to adequately test the model for $K(z)$, our derivations of $K_{max}$ and $H_K$ as functions of the static stability, where $K_{max} \propto (\frac{z_v}{L})^{-0.6}$ and $H_K \propto (\frac{z_v}{L})^{0.6}$, agree well with the expected dependencies of the two parameters on the background potential temperature lapse rate ($\gamma$). In the original model (Parmhed et al.,

2004), keeping $H_K$ constant gives $K_{max} \propto \gamma^{-0.5}$, while keeping $K_{max}$ constant gives $H_k \propto \gamma^{0.5}$. Thus, the exponent in our empirical power-law relation is similar to the theoretical exponent in the model's power-law relation. While these results are promising in terms of applicability of $K_{Int}$-approach in determining turbulent heat fluxes, it remains to be investigated whether the local M-O stability metric ($\frac{z_{v,t}}{L}$) is a good proxy for $\gamma$. As a first-order estimate, working with



the assumption that $\gamma$ is inversely proportional to the incoming longwave radiation at the surface ($L_{down}$), we performed a correlation analysis between ($\frac{z_v}{L}$) and $L_{down}$ timeseries. While there is no statistically significant negative correlation for the 30-min values, there is a significant negative correlation when the smoothed timeseries of $\frac{z_v}{L}$ are used. In other words, the long term (multi-day) fluctuations in the static stability are shown to be negatively correlated with the long

term $L_{down}$ fluctuations (results not shown). Considering the successful performance of the $K_{Int}$-approach based on the assumed linear-Gaussian profile for $K(z)$, a possible extension of our work would be to investigate whether a more exact parametrization for $K$ (as a function of $z$, $\frac{\partial U}{\partial z}$, and $\frac{\partial T}{\partial z}$) in the surface layer can be determined as a solution to the governing equations (heat and momentum balance) for a katabatic flow. On the other hand, the poor performance of $K_{Int}$-approach is assessing the sensible heat fluxes, reveals a shortcoming in the assumption that the eddy diffusivity

($K_H$) and eddy viscosity ($K_M$) share the same parametrization. One way to improve for this shortcoming would be to introduce a vertically varying eddy Prandtl number, $Pr(z)$.

## 5   Conclusions

The main objective of the study was to evaluate commonly used bulk approaches for simulating turbulent heat fluxes at a sloped glacier surface. In particular, we investigated the stability components of the bulk methods, and attempted to improve

upon these, assuming site-specific roughness length values are available. In addition to the K-approach with different parameterizations for the bulk exchange coefficient ($C$), we included a set of less commonly used bulk approaches developed from katabatic flow models (Grisogono and Oerlemans, 2001; Oerlemans and Grisogono, 2002). These 23 bulk schemes, for simulating 30-min turbulent fluxes of momentum, sensible and latent heat, were evaluated against the equivalent fluxes obtained by an open path eddy-covariance (OPEC) method. The evaluation was performed at a point scale of an alpine glacier in BC, using

one-level meteorological and OPEC observations from a multi-day period in the 2010 and 2012 summer seasons. Prior to the inter-comparisons, the OPEC data were subjected to a set of quality control corrections and filters. Few 30-min data segments met the criteria for the near-neutral static stability and the steady-state turbulence conditions, revealing a challenge to perform this type of analysis at sites dominated by the stable conditions and drainage (katabatic) flows. Despite the small OPEC data sample available for determining the surface roughness lengths ($z_{0v,0t,0q}$) and the high-quality 30-min turbulent fluxes, the

evaluation results derived from the two independent field seasons agree. A summary of our main findings are:

        Bulk exchange coefficients could be derived from the OPEC-derived roughness lengths for neutral stratification, where the mean log values for $z_{0v,0t,0q}$ over the entire period are assumed representative of the actual surface roughness and constant in time. In other words, the widely used K-approach facilitated by the standard meteorological observations on glaciers, provides a good approximation of the OPEC-derived 30-min turbulent fluxes during the neutral stability condi-

tions. However, as the stability increases, the K-approach with commonly used parameterizations of the bulk exchange coefficients performs relatively poorly in comparison to the bulk approaches based on a katabatic flow model.

        According to the Monin-Obukhov (M-O) stability theory, under stable stratification, the bulk exchange coefficient should be modified by the universal stability functions. However, the stability functions widely used in glacier studies perform





relatively poorly on our site. The K-approach with commonly used parameterizations for $C$, with or without the stability corrections, overestimates the sensible heat fluxes ($Q_H$) over the observational period. The overestimation is due to a positive mean bias in the modeled friction velocities ($u_*$). Thus, without an adequate model for $u_*$, the use of the stability corrections for $Q_H$ will probably fail. Considering that similar findings are derived from a different glacier in this region (Fitzpatrick et al., 2017) these conclusions might apply to other alpine glaciers in similar climatic settings.

The inability of M-O stability functions to adequately correct for the overestimation of $u_*$ and $Q_H$ during stable conditions was partly resolved by developing the new empirical stability corrections tuned to our data. These new stability functions work well when OPEC-derived $\frac{z_{v,t}}{L}$ is used as an input to the bulk method. If OPEC data is not available, however, $\frac{z_{v,t}}{L}$ cannot be successfully predicted, and therefore the new stability functions fail to improve the bulk model performance. The proposed fixed-point iterative method for predicting $\frac{z_{v,t}}{L}$, which has been previously argued to perform successfully (Berkowicz and Prahm, 1982; Lee, 1986; Munro, 1989), fails to provide a good approximation of OPEC-derived $\frac{z_{v,t}}{L}$.

Getting a correct estimate of $u_*$, rather than a correct estimate of $\frac{z_{v,t}}{L}$, is shown to be more important for improving the simulation of $Q_H$. We found that, contrary to the K-approach predictions, the relation between the OPEC-derived 30-min $u_*$ and observed wind speed ($U_z$) is not well represented by a linear fit. Our data, instead, reveals $u_* \propto \sqrt{U_z}$, where the constant of proportionality is a nonlinear function of $\frac{z_{v,t}}{L}$.

The bulk approach derived from the katabatic flow model, abbreviated as $K_{Int}$ approach, outperforms any other bulk scheme in simulating $u_*$, because the $K_{Int}$ approach is rooted in $u_* \propto \sqrt{U_z}$. The approach performs poorly in simulating $Q_H$, however, revealing a shortcoming in the commonly used assumption that eddy viscosity and diffusivity can use the same parametrization. Applying the $K_{Int}$ approach to assess $u_*$, which is then used in the K-approach with the new M-O stability function to assess $Q_H$ gives the best performance across all the bulk methods we tested. In short, $Q_H$ can be more successfully modeled with $Q_H \propto u_* \, \Delta T$, where the constant of proportionality is a function of the M-O stability parameter, rather than using the K-approach that relies on $Q_H \propto U_z \, \Delta T$.

The bulk exchange coefficient in the $K_{Int}$ approach is found to be a function of $\frac{z_{v,t}}{L}$, in a very similar way as the bulk exchange coefficient is a function of the background potential temperature lapse rate ($\gamma$) in the original model of Grisogono and Oerlemans (2001). As expected by the original model and confirmed by our results, the atmospheric stability increases (larger $\gamma$ in the model or larger $\frac{z_{v,t}}{L}$ in our data) as the bulk exchange coefficient decreases. Provided that $\gamma$, taken from observations or regional climate models, is shown to be a good proxy for $\frac{z_{v,t}}{L}$, the $K_{Int}$ approach could be a predictive tool for simulating the turbulent fluxes during prevailing katabatic flows. Clearly more observations are needed to adequately evaluate the model, in particular, multi-level observations of wind speed and temperature within the boundary layer above the glacier surface.

As initially stated, this study provides a foundation towards developing a better parameterization of turbulent fluxes on sloping glacier surfaces subject to katabatic flows. OPEC data from an expanded SEB monitoring program on glaciers in the



region, commenced in 2014, will allow for further analysis and help us to address the outstanding questions that arose in this study. One of the biggest challenges, as we see it, for both measuring and parameterizing the turbulent heat fluxes, is the presence of the intermittent turbulence that often accompanies katabatic flows. To tackle the challenge from the measurement side one would need to use the detection methods of intermittency through a spectrum and/or wavelet analysis on the eddy-

5  covariance data. On the modeling side, one would need to invest into new ways of parameterizing the eddy viscosity that go beyond the K-approach and beyond the steady-state governing equations for katabatic flow.

*Author contributions.* VR participated in the 2012 field campaign, developed and performed the analysis, and wrote the initial version of the manuscript. BM provided financial and field support and contributed to manuscript preparation. JS led the 2010 field campaign and contributed to the manuscript refinement. NF processed the OPEC data and contributed to the manuscript refinement. MAT contributed to

10  the method development. SJD provided logistical support, meteorological data and contributed to the manuscript refinement.

*Acknowledgements.* Funding supporting this study was provided through the Natural Sciences and Engineering Research Council of Canada (Discovery grants to V. Radić, B. Menounos, S.J. Déry) and Canada Research Chairs Program. Science Horizons are thanked for additional financial support. Theo Mlynowski is thanked for the field setup in 2010, and Andrew Duncan for the field support in 2012. Branko Grisogono, Christian Schoof and Elisa Mantelli are thanked for discussions and constructive criticism.



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
