# Peer review of "EVALUATION OF DIFFERENT METHODS TO MODEL NEAR-SURFACE TURBULENT FLUXES FOR A MOUNTAIN GLACIER IN THE CARIBOO MOUNTAINS, BC, CANADA"

_The Cryosphere, 2017_

## Referee Comment (RC1) · J.P. Conway (Referee) · 12 Jul 2017

**Referee review for manuscript tc-2017-80, under review for The Cryosphere.**

"EVALUATION OF DIFFERENT METHODS TO MODEL NEAR-SURFACE TURBULENT FLUXES FOR AN ALPINE GLACIER IN THE CARIBOO MOUNTAINS, BC, CANADA" by Valentina Radic, Brian Menounos, Joseph Shea, Noel Fitzpatrick, Mekdes A. Tessema, and Stephen J. Déry.

**Reviewer: J Conway**

**General Comments**

This manuscript addresses the turbulent fluxes of sensible ($H$) and latent heat ($LE$) over small mountain glaciers using in-situ measurements and theoretical models. The topic is not only extremely relevant to efforts to resolve cryosphere-climate relationships, but has many unresolved questions. The manuscript is novel in that it presents new measurements that add to the relatively small body of literature on turbulent fluxes in these environments. The roughness length parameter used in the most common Monin-Obukhov (M-O) bulk approaches to modelling these fluxes is successfully derived from measured fluxes in ideal neutral-stability conditions. The derived values agree well with other estimates from the literature, with the roughness length for momentum being 2 order of magnitude larger than that for temperature. The authors also derive the roughness length for humidity – a seldom performed activity – and show it to be of the same order as temperature. The authors then delve into modelling $H$ with the M-O bulk method, showing a general overestimation of $H$ and the friction velocity ($u^*$). The authors test various methods to represent $H$ and $u^*$ using a combination of measured and modelled turbulence variables within a variety of theoretical frameworks. Some of these give promising results, but overall, no theoretical framework is markedly better at modelling $H$ using only mean wind speed and temperature at one height above the glacier surface. The best method presented is the simplest, relating $H$ to a quadratic of the 2-meter air temperature (Oerlemans and Grisogono, 2002). The measured values of $u^*$ and the M-O stability parameter ($z/L$) are shown to be poorly predicted by bulk models. If $u^*$ is known, then $H$ can be predicted well by M-O bulk schemes.

Many useful analyses are presented, and the paper should be of interest to researchers modelling the surface energy and mass balance of mountain glaciers. However, several areas need to be addressed if the most important results for this community are to be highlighted. In particular, more focus needs to be made on methods that can be implemented solely with mean wind, temperature and humidity, otherwise the results would be more suited to a meteorological journal addressing the underlying theoretical framework for the schemes. Several of results discussed contain ambiguities and potential spurious self-correlation (such as the relationship between $u^*$ and the newly derived stability functions). These need a much fuller discussion elsewhere, and distract from the main thrust of the paper. That being said, I do think the analysis of turbulence data produces some key results (e.g. that $u^*$ does not relate well to mean wind speed and that $z/L$ is poorly predicted by bulk models), which could be

presented more explicitly. This could take the form of a dedicated comparison of $z/L$ with that predicted by M-O, correlation with bulk Richardson stability parameter etc. These results would better set the context for the performance of each parameterisation. One other key result of the paper needs to be further highlighted - the good performance of the simple katabatic model compared to the bulk model.

The large and somewhat overwhelming quantity of analysis makes the manuscript hard to follow at times. Many of the panels in the later figures (Fig. 9, 10 and 12) deserve to be their own figure as they address a distinct point from other panels. As discussed above, I would prefer to see a dedicated examination of the relationships between $U$, $u^*$, $H$, $z/L$, rather than the dispersed results at present. This would help the reader to evaluate different assumptions within the theoretical framework, and point more convincing toward new theories. At present, there is a danger that readers are confused by the various ways in which the eddy covariance data are used.

Most of the analyses are clear, appropriate and well supported by theory. The analyses of the new stability function and $K_{int}$, however, introduce another layer of theoretical framework that deserves further analysis and discussion. Some of this is beyond the measurements available to the authors (i.e. profile measurements) and could be beyond the scope of the journal. Several aspects of these are quite uncertain and need more discussion. This includes the derivation of $K_{max}$ and $H_k$ – does the variation of one, when the other is held constant, reveal the inadequacy of the method? The dependence shown in Fig. 12 (d) and (e) is between $z/L$ and $u^*/U$ – and as $z/L$ contains u*, thus there is potential for spurious self-correlation.

In regard to the calculation of new stability function – to properly address this requires profile measurements, otherwise it is simply a circular way to use the measured fluxes to correct the bulk method. It appears that as stability increases, momentum flux decreases while $H$ does not – this points to the influence of a wind speed maximum, where $u^*$ goes to zero at close the height of the maximum, which will be lower for lower wind speed (Denby and Greuell, 2000). The authors need to reflect on the sensor heights in regard to flux-profile relationships in presence of katabatic, and how these may be affecting the observed relationships between $u^*$, $z/L$ and $H$.

The authors need to be careful that the key results outlined in the abstract and conclusions are explicitly analysed in the paper. At present, there is some support for the alternate parameterisation schemes presented here, but they depend too much on in-situ turbulence measurements to be used widely. These results are still worth presenting, they just need to be more thoroughly analysed (perhaps elsewhere) before definitive statements can be made. It is good to see movements toward developing new turbulent heat flux parameterisations for mountain glacier environments, which is an essential step for the community.

In summary, the manuscript should make a good contribution to the literature on this subject if a number of issues are addressed.

**Specific Comments (**page-line**):**

1-8: "the bulk method" – please clarify what is meant by this term. In general, the terminology used needs clarification. The term K-approach is not likely to be familiar for most readers of *The Cryosphere*, and is easily confused with the $K_{Int}$ approach introduced later. Consider using a different term here to distinguish the bulk methods in which $K$ scales with $u^*$, perhaps "M-O bulk schemes".

1-12: "The OPEC-derived 30-min momentum flux is linearly related to the measured wind speed, contrary to the proposed quadratic relation by the commonly used bulk methods." – This result is not shown but rather hinted at (26-4). Needs to be clearly analysed within the paper for this statement to be supported in the abstract.

1-15: "In agreement with the katabatic flow model, we show that in a more stable atmosphere the bulk exchange coefficient for momentum is smaller." Again, the relationship is not analyses specifically, so it is hard to see this a key result. Please revise.

1-16: "The sensible heat flux can be more successfully modeled if the bulk exchange coefficients for momentum and heat are allowed to follow different parametrization schemes, rather than assuming equal schemes as is the case in the common bulk methods." But the data don't seem to show a large improvement for the more complicated schemes when only mean wind speed and temperature are used. These schemes often rely on measurements of $z/L$ so aren't easily transposed in space and time. Please revise.

2-15: References needed here.

2-21: Please consider adding Guo, X., Yang, K., Zhao, L., Yang, W., Li, S., Zhu, M., Yao, T., and Chen, Y.: Critical Evaluation of Scalar Roughness Length Parametrizations Over a Melting Valley Glacier, Boundary-Layer Meteorology, 139, 307-332, 2011.

3-10: "valley glaciers" – do you mean mountain glaciers? Also, it could be worth consistently referring to mountain or alpine glaciers if the two are to be treated similarly (see 4-26 & 4-27).

3-12: *z0v* is a mathematical variable that relates the flux and the gradient, and, as such, is not always related to the turbulence generated by roughness elements at the surface. Outer-layer turbulence can, for instance, increase the momentum flux in the surface layer, thereby changing *z0v*, while the surface roughness elements remain constant. Please revise.

5-10: Please provide model numbers of the instruments.

6-13: Was the sonic temperature corrected for the effects of water vapour?

7-4: WPL corrects for fluctuations in the water vapour density induced by high-frequency changes air temperature, so is associated with turbulent eddies. Please revise.

7-28: The assumption of a melting surface, even on temperate glaciers, does not always hold during the melt season. It would be better to use the SEB, or an air temperature threshold to screen periods in which the surface is likely to be melting to validate the use of this assumption. One period in September 2012 shows air temperature well below freezing, so would almost definitely have lower surface temperature. Also, periods in late August and early September 2012 have air temperature close to 0°C, so the surface temperature is likely to be less than 0°C during these periods. Please discuss further and consider filtering bulk-method results based on periods in which high confidence can be had in the melting surface assumption.

8-1: Assuming a melting surface in preference to outgoing longwave radiation measurements or SEB closure is predicated on there always being ample energy available for melt. Where this is not the case, SEB closure can give much better results than the assumption of the melting surface (e.g. Conway and Cullen, 2013). I agree that when the surface is most definitely melting, then making the assumption of 0°C is a good way to remove uncertainty in the calculation of surface temperature. Please revise this statement.

8-25: Please refer to the later choice to extend the $z/L$ range to $z/L = 2$ (11-13).

10-5: It would be useful to introduce the filters used to select the 30-min periods used to compare fluxes.

10-7: The term K-approach is introduced with no background. Is there a less ambiguous term to use for this family of bulk approaches, given that none of the acronyms include K, and you introduce $K_{Int}$ later as a separate method?

12-7: Why were errors associated with the air temperature not included in the error analysis?

15-8: Why were low wind speed periods not included in the analysis? It would be more appropriate to only include the stationarity and wind direction filters, as the other filters are specifically designed for retrieving roughness lengths, rather than removing bad flux estimates. I would expect this to change the comparison significantly, especially the inclusion of low wind speed periods.

23-16: The static stability referred to in the OG model is that of the background airmass. As we have no information on the background stability we don't know if this is necessarily reflected in the stability of the surface layer ($z/L$). Please revise.

26-14: Is there a way to evaluate $K_{int}/K_{max}$ without measured $z/L$? For consistency, it would be useful to discuss if this is possible.

27-9: Why are those with measured $z/L$ but not those with measured $u^*$ included? I think it would be better to only present schemes that do not use any time-varying information from

the OPEC system, as these are the parameterisations that are of use to those wishing to use the bulk method.

29-15: It is unclear which filtered periods were used here. Please clarify.

29-27: There is a need to discuss potential systematic biases in both surface and air temperature and how these could propagate into the calculation of $H$. This could include additional screen using the SEB to identify melting periods only to compare to OPEC fluxes, and/or a comparison of sonic-temperature with the unventilated air temperature measurements or application of corrections for low wind speed (Huwald et al. 2009) .

31-7: "fails to successfully simulate $QH$." I would be careful making this statement, as you could argue that it does simulate $H$ fairly well, not just as well as some other, more site-specific schemes. The main failing of the bulk schemes presented here, is the failure to model $u^*$. Please revise.

31-8: "Note that the new stability correction acts in the opposite direction than those commonly used for glacier studies: in our case, the modeled QH needs to increase, rather than be suppressed, as the stability increases." Yes, but only because $u^*$ is overestimated. $H$ still decreases relative to stable conditions. Thus, the result is not so much about the effect of atmospheric stability, but the trouble with specifying turbulence in the presence of katabatic flow. Please revise.

33-4: This paper is not available to the reader at the present time, so it is hard to assess this statement.

33-13 to 16: This is a key result and should feature more highly in the manuscript.

33-22: "Applying the $K_{Int}$ approach to assess $u^*$, which is then used in the K-approach with the newM-O stability function to assess $QH$ gives the best performance across all the bulk methods we tested". Yes, but the fit between $u^*$, $U_z$ and $K_{int}$ is informed by measurements of $z/L$ (which contains $H$) as is the fit between $u^*$, $z/L$ and $H$ in the stability function, so it is not surprising that this function works the best.  Please discuss the self-correlation and revise.

Figures 7 and 11: Consider including the numbers assigned to each parameterisation above the columns of each figure panel to aid the reader.

Figure 12: The order of parameterisations in panel (a) needs to be consistent with Table 3 – i.e. the first parameterisations introduced at the top. As with Fig 7, needs to have the numbers assigned to the schemes next to the y-axis labels.

Table 1: Please include the units for the roughness lengths in here.

Table 3: Consider removing parameterisations 17, 19, 21 and 23 as they are essentially duplicates of 16, 18, 20 and 22. Also consider adding lines between the sub-sets of parameterisations.

**Editorial Comments**

2-9: "recourses" -> "resources"

4-28: "is monitored" -> "has been monitored"

4-31: -> "In the glacier vicinity, two year-round automatic weather stations have been in operation since 2007/2008 (Déry et al., 2010). The stations are situated on the lateral and terminal moraines, and are referred to as AWSup …. And AWSlow…. respectively."

5-12: Do you mean AWSlow?

7-5: "potentially high and low frequency loss," -> "potential loss of high and low frequency signal"

17-3: -> "Obukhov length ($L$)"

23-7: -> "for 2010 and 2012" or "for 2012 and 2010".

26-9: -> "best estimate of the friction velocity or momentum flux among all the bulk schemes we tested so far (compare Fig. 11 to Fig. 5)." Need to help the reader to navigate between the results.

26-23: It would be consistent with the presentation of other schemes to include (22) and (23) before (18) and (19), respectively. i.e. first using iterative $z/L$, then using measured. The same for (13) and (14).

26-23: "empirical function for $K_{max}$" and "empirical function for $H_k$" - please refer to equations 35 and 36 here.

27-1: -> "hybrid methods drops (Table 3)."

30-34: -> "resolve issue (i)"

33-3: -> "modeled friction velocity"

---

## Referee Comment (RC2) · R. Dadic (Referee) · 7 Aug 2017

**Evaluation of Different Methods to Model Near-Surface Turbulent Fluxes for an Alpine Glacier in the Cariboo Mountains, BC, Canada**

August 7, 2017

Dear Editor,

I reviewed the paper "Evaluation of Different Methods to Model Near-Surface Turbulent Fluxes for an Alpine Glacier in the Cariboo Mountains, BC, Canada" Ice by Radic and others.

The paper discusses the often neglected topic of the accuracy of turbulent fluxes in mountain environments. The calculations of turbulent fluxes are addressed using data as well as a range of relevant parameterisations and the results are well worth publishing. I look forward to seeing this important work published.

Generally the paper is well written, but I did find it cumbersome to read, because it includes many different parameterisations that are not easily distinguishable in the text. So I suggest that the authors consider a restructuring of the methods to clarify the difference between the model runs they performed and maybe "cluster" the methods that are similar.

I discuss my other suggestions to the authors in the comments below.

Sincerely,
Ruzica Dadic

**General comments**

- My main concern with the paper is that it neglects the very stable conditions by only looking at conditions where wind speed is ¿3m/s or (the moisture/temperature gradients are large enough). I appreciate that the measurement of turbulent fluxes under very stable conditions are harder to obtain because the mean flow is non-stationary and characterised by brief episodes of intermittent turbulence Mahrt [1989]; Beljaars and Holtslag [1991]; Mahrt [1998]; Cheng et al. [2005] . Considering the significant amount of the periods where low wind speeds occur (Figure 4 in the submitted manuscript), those periods should not be neglected when trying to improve the turbulent fluxes parameterisations over glaciers. A number of studies have been dedicated to finding valid flux-profile relationships for very stable conditions, such as are often found over snow and ice surfaces [e.g. Webb, 1970; Kondo et al., 1978; Lettau, 1979; Brutsaert, 1982; Holtslag and de Bruin, 1988; Beljaars and Holtslag, 1991; Cheng and Brutsaert, 2005; Grachev et al., 2007] and those studies have also been applied to snow and ice surfaces [Pomeroy et al., 1998; Jordan et al., 1999; Sharan, 2009; Dadic et al., 2011].

- All bulk methods assume a logarithmic profile, and they only differ in what stability correction they use. This should be clarified in the manuscript.

- Figures 1, 2, and some of Figure 4 (radiation, precipitation, wind direction) are not needed in this paper and can be removed.

- Figure 5: It is of no surprise that pretty much all 4 methods in this Figure have the same results, considering they all use he bulk method at almost neutral conditions. By neglecting the stable conditions, they don't have much reason not to vary. I am therefore not sure what the point of his comparison is.

- P18–19: It is not surprising that the "parameterisations" which use measured $u_*$ as input lead to an increase in fit with the data. $u_*$ goes into the $Q_E$ equation by the power of 4, it's proportional to $Q_H$. It changes $L$ with the power of 3, so will disproportionally decrease $z/L$. Some of this discussion (why $u_*$ has more influence on the turbulent fluxes calculation than $z/L$) might be easier to understand by just looking at the equations and the relevance of the different parameters.

  Furthermore, I the observation on page 18 (L1–3) that the $C_{log}$ and $C_S R$ methods are not justified in table 3, where the difference between the $u_*$ models in the correlation coefficient $r$ is between 0.94 and 0.95 for $Q_E$ and between 0.82 and 0.85 for $Q_H$, which is not exactly significant. I am not sure how to address this problem, but I'm sure the authors can come up with more robust conclusions than that.

- p 29, L1-2: Considering that the authors have most SEB components to actually calculate the surface temperature, and that the surface temperature is an important feedback for the TF, the authors should consider calculating the surface temperature and including it in their calculations using the different paramaeerizations. It would be interesting what effect the different parameterizations have on surface temperature. I do not expect the authors to change all their results now, but maybe it's worth a discussion in the paper.

- p 30, L26-30: Considering that only near-neutral conditions are used for this study, I am not surprised that the stability corrections show very little difference when modelling the fluxes.

- P31, L11-13: As far as I remember, the reason why the turbulent fluxes are suppressed in Conway and Cullen (2013) is that they assumed the log-linear relationship to be valid under very stable conditions. The log-linear relations, however, do not allow for significant fluxes to occur at very strong stability[Monin and Yaglom, 1971; Mahrt, 1998; Pleim, 2006] and underestimate the turbulent fluxes over these conditions [e.g. Deardorff, 1968; Webb, 1970; Kondo et al., 1978; Louis, 1979; Hogstrom, 1988; Launiainen, 1995; Mahrt, 1998; Jordan et al., 1999; Stössel et al., 2010].

  The Figure below is unpublished, but it shows how the log-linear profile behaves under very stable conditions ($\zeta > 1$ or low wind speeds) for different climatic conditions.

**References**

Beljaars, A. and Holtslag, A. (1991). Flux parameterization over land surfaecs for atmospheric models. *Journal of Applied Meteorology*, 30:327–341.

Brutsaert, W. (1982). *Evaporation into the Atmosphere: theory, history and applications*. D. Reidel Publishing Co.

Cheng, Y. and Brutsaert, W. (2005). Flux-profile relationships for wind speed and temperature in the stable atmospheric boundary layer. *Boundary–Layer Meteorology*, 114:519–538.

Cheng, Y., Parlange, M., and Brutsaert, W. (2005). Pathology of Monin-Obukhov similarity in the stable boundary layer. *Journal of Geophysical Research*, 110(D6).

[Figure]

Figure 1: Turbulent fluxes as function of wind speed (1–5 ms$^{-1}$) for 9 different climatic conditions calculated with 5 different flux-profile relationships for the stability correction. The Basinger-Dyer parameterisation is purely log-linear and surpasses turbulent fluxes when $\zeta > 1$.

Dadic, R., Mott, R., Lehning, M., Carenzo, M., Anderson, B., and Mackintosh, A. (2011). Sensitivity of turbulent fluxes to wind speed over snow surfaces in different climatic settings. *Advances in Water Ressources*, submitted.

Deardorff, J. (1968). Dependence of air-sea transfer coeffinients on bulk stability. *Journal of Geophysical Research*, 73(8):2549–2557.

Grachev, A., Andreas, E., Fairall, C., Guest, P., and Persson, P. (2007). SHEBA flux-profile relationships in the stable atmospheric boundary layer. *Boundary–Layer Meteorology*, 124:315–333.

Hogstrom, U. (1988). Non-dimensional wind and temperature profiles in the atmospheric surface layer: a re-evaluation. *Boundary–Layer Meteorology*, 42:55–78.

Holtslag, A. and de Bruin, H. (1988). Applied modeling of the nighttime

surface energy balance over land. *Journal of Applied Meteorology*, 27:689–704.

Jordan, R., Andreas, E., and Makshtas, A. (1999). Heat budget of snow covered sea ice at north pole 4. *Journal of Geophysical Research*, 104(C4):7785–7806.

Kondo, J., Kanechika, O., and Yasuda, N. (1978). Heat and momentum transfers under strong stability in th eatmospheric surface layer. *Journal of the Atmospheric Sciences*, 35:1012–1021.

Launiainen, J. (1995). Derivation of the relationship between the obukhov stability oarameter and the bulk richardson number for flux-profile studies. *Boundary–Layer Meteorology*, 76:165–179.

Lettau, H. (1979). Wind and temperature profile prediction for diabatic surface layers including strong inversion cases. *Boundary–Layer Meteorology*, 17:443–464.

Louis, J.-F. (1979). A parametric model of vertical eddy fluxes in the atmosphere. *Boundary–Layer Meteorology*, 17:187–202.

Mahrt, L. (1989). Intermittency of the atmospheric boundary layer. *J. Atmos. Sci.*, 46:79–95.

Mahrt, L. (1998). Stratified atmospheric boundary layers and breakdown of models. *Theoret. Comput. Fluid Dynamics*, 11:263–279.

Monin, A. and Yaglom, A. (1971). *Statistical Fluid Mechanics: mechanics of turbulence, Vol. 1*. Number 769 pp. MIT Press, Cambridge, Massachusetts.

Pleim, J. (2006). A simple, efficient solution of flux-profile relationships in the atmospheric surface layer. *Journal of Applied Meteorology and Climatology*, 45:341–347.

Pomeroy, J. W., Gray, D. M., Shook, K. R., Toth, B., Essery, R. L. H., Pietroniro, A., and Hedstrom, N. (1998). An evaluation of snow accumulation and ablation processes for land surface modelling. *Hydrological Processes*, 12:2339–2367.

Sharan, M. (2009). performance of various similarity functions for nondimensional wind and temperature profiles in the surface layer in stable conditions. *Atmospheric Research*, 94:246–253.

Stössel, F., Guala, M., Fierz, C., Manes, C., and Lehning, M. (2010). Micrometeorological and morphological observations of surface hoar dynamics on a mountain snow cover. *Water Resources Reasearch*, 46(W04511).

Webb, E. (1970). Profile relationships: The log-linear range, and extension to strong stability. *Quarterly Journal of the Royal Meteorological Society*, 96:67–90.

---

## Author Response (AR1)

**Dear Dr. Thomas Mölg,**

Please find attached our responses to all reviewers' comments and the revised manuscript. We thank both reviewers for their detailed review that helped us to significantly improve the manuscript. Please note that due to substantial reorganization of the text in the revised manuscript (mainly streamlining of methods and results, following the reviewers' comments) we do not attach a draft with the tracked changes. Instead, we summarize the major changes in the revised manuscript that incorporate comments from both referees:

1) Restructured (streamlined) the methods and results sections. In particular, all evaluated bulk methods in the study are clustered into two main types: C-methods and methods based on katabatic models. C-methods consist of four subgroups: C_log, C_Rib, C_M-O and C_SR, whereas the second cluster consists of the C_kat method and K_Int method. All the bulk methods depend on mean meteorological variables only (i.e. there is no application of OPEC data in the bulk methods). All other variants of the bulk methods (i.e. those with OPEC-derived variables) are now presented as part of the sensitivity analysis;

2) Removed the section on newly derived stability functions (following the comments from Referee #1);

3) Improved the discussion on possible spurious self-correlations and removing the results where the self-correlation was present;

4) Introduced a proxy variable for the background temperature lapse rate by using the near-surface air temperature observations from the two nearby meteorological stations, at different altitudes and in the glacier vicinity, in addition to the temperature measurements from the glacier station. The usage of these data led us to identify a dependency of parameters in the K_Int method on the proxy variable (i.e. difference in off-glacier and on-glacier near-surface air temperature). This empirical relationship allowed us to assess the parameters purely from the mean meteorological variables, instead of using the OPEC-derived stability parameter (z/L) as was the case in the initial manuscript;

5) Quantified the errors in the modelled sensible heat fluxes resulting from the radiative overheating of the temperature sensor;

6) All results (sensible and latent heat fluxes) are now evaluated for the cases with 30-min wind speed exceeding 1 m/s (instead of wind speed exceeding 3 m/s as was the case in the initial manuscript); and

7) Revised several figures and added a few that explicitly show dependencies among variables: z/L, Bulk Richardson number, wind speed and air-surface temperature difference.

Please find below our responses to each referee. The responses are given in bold font, while any quotes from the revised manuscript are copied in italics.

We thank you sincerely for handling our manuscript and look forward to your decision.

Best regards,
Valentina Radic

**Responses to Referee #1's comments:**

● Many useful analyses are presented, and the paper should be of interest to researchers modelling the surface energy and mass balance of mountain glaciers. However, several areas need to be addressed if the most important results for this community are to be highlighted.
In particular, more focus needs to be made on methods that can be implemented solely with mean wind, temperature and humidity, otherwise the results would be more suited to a meteorological journal addressing the underlying theoretical framework for the schemes. Several of results discussed contain ambiguities and potential spurious self-correlation (such as the relationship between u* and the newly derived stability functions). These need a much fuller discussion elsewhere, and distract from the main thrust of the paper. That being said, I do think the analysis of turbulence data produces some key results (e.g. that u* does not relate well to mean wind speed and that z/L is poorly predicted by bulk models), which could be presented more explicitly. This could take the form of a dedicated comparison of z/L with that predicted by M-O, correlation with bulk Richardson stability parameter etc. These results would better set the context for the performance of each parameterisation. One other key result of the paper needs to be further highlighted - the good performance of the simple katabatic model compared to the bulk model.

**We appreciate the constructive criticism aimed at tailoring our presentation of results to the glaciological community readership. We thus revised the manuscript by highlighting more explicitly the key findings related to the bulk methods that can be implemented solely with mean wind, temperature and humidity. Also, we now explicitly present the relation among measured u\*, U, z/L, H and the bulk Richardson stability parameter (see the figures below). As indicated by the referee, we hope that, by explicitly presenting these relations, we improve the interpretation of the bulk methods' performance.**

[Figure]

*Figure 5: Bias (modeled minus observed values) in $u\_*$ and $Q\_H$ versus the wind speed. Red points show values during the neutral conditions ($|\frac{z_{v,t}}{L}|<$0.1) only, and blue points indicate values during the prevailing katabatic flow.*

[Figure]

*Figure 7: Modeled z/L with the fixed-point iterative scheme in the $\mathrm{C_{M-O}}$ method and the Bulk Richardson number ($R_{ib}$) against the OPEC-derived stability parameter (z/L obs). Dashed black line shows 1:1 line. Also shown is a dependency of the 30-min OPEC-derived stability parameter (z/L obs) on the wind speed ($U_z$) and the near-surface air temperature ($T_z$).*

● The large and somewhat overwhelming quantity of analysis makes the manuscript hard to follow at times. Many of the panels in the later figures (Fig. 9, 10 and 12) deserve to be their own figure as they address a distinct point from other panels. As discussed above, I would prefer to see a dedicated examination of the relationships between U, u*, H, z/L, rather than the dispersed results at present. This would help the reader to evaluate different assumptions within the theoretical framework, and point more convincing toward new theories. At present, there is a danger that readers are confused by the various ways in which the eddy covariance data are used.

**We agree on the overwhelming appearance of the analysis and, following the referee's suggestions, we streamlined the results (e.g. put more focus on explicitly showing the relationships among U, u*, H, and z/L) and reorganized the paper to focus on what is most important. We also clustered the bulk methods into two main categories (where only mean variables are used in the method), addressing the remaining bulk methods (with OPEC-derived variables) mainly as a part of a sensitivity analysis.**

● Most of the analyses are clear, appropriate and well supported by theory. The analyses of the new stability function and Kint, however, introduce another layer of theoretical framework that deserves further analysis and discussion. Some of this is beyond the measurements available to the authors (i.e. profile measurements) and could be beyond the scope of the journal. Several aspects of these are quite uncertain and need more discussion. This includes the derivation of Kmax and Hk – does the variation of one, when the other is held constant, reveal the inadequacy of the method? The dependence shown in Fig. 12 (d) and (e) is between z/L and u*/U – and as z/L contains u*, thus there is potential for spurious self-correlation.

**We agree that a more detailed analysis regarding the Kint method in assessing the turbulent fluxes is beyond the scope of this study, mainly because of the limited amount of data on this site. We make a more clear distinction in the revised manuscript between the findings fully supported by our data and the findings which are partly supported but insightful enough to trigger further research when/if the data become available. We now removed the analysis with the self-correlation. Instead, in the revised manuscript, we identified a better way to relate the parameters in K_Int method with the use of near-surface air temperature measurements from**

**the two stations in the glacier's vicinity.**

● In regard to the calculation of new stability function – to properly address this requires profile measurements, otherwise it is simply a circular way to use the measured fluxes to correct the bulk method. It appears that as stability increases, momentum flux decreases while H does not – this points to the influence of a wind speed maximum, where u* goes to zero at close the height of the maximum, which will be lower for lower wind speed (Denby and Greuell, 2000). The authors need to reflect on the sensor heights in regard to flux-profile relationships in presence of katabatic, and how these may be affecting the observed relationships between u*, z/L and H.

**We agree that, if one is to derive new stability functions to be used for glacier studies, more measurements (in particular, profile measurements) are needed. Our initial goal was not to derive the new stability functions, but to compare the performance of the optimized stability function (one that assures the best match between modelled and observed fluxes) with the performance of the stability functions commonly used in the glacier studies. However, considering the length of the manuscript and the fact that more observations are needed to adequately analyze the effect of stability, we now removed this section. Instead, we extended the discussion, as suggested by the referee, on the potential influence of a wind speed maximum on the bulk method performance.**
***Page 22/23 in the revised manuscript (copied here in the Latex form):***
*While the original C-methods, however, overestimate $Q_H$ during the katabatic conditions, the C-methods with stability corrections and measured $u_*$ underestimate the fluxes (Table \ref{tab:evaluation results} and Fig.\ \ref{fig: scatter plot with sensitivity tests}). The overestimation of $Q_H$ during katabatic flows has also been shown in \cite{Denby_Greuell2000} and explained by a failure of M-O theory in the presence of shallow katabatic wind speed maximum. At the wind speed maximum, measured $u_*$ approaches zero, while the C-method assumes constant momentum flux in the surface layer and therefore overestimates $u_*$. The overestimation is less pronounced for $Q_H$ than for $u_*$ because the reduced turbulence at the wind speed maximum leads to an increase in the air-surface temperature difference, and subsequently an increase in the measured $Q_H$. However, when measured $u_*$ is used in the $\mathrm{C_{M-O}}$ or $\mathrm{C_{Rib}}$ method, assuming that the eddy diffusivity is as effective as eddy viscosity ($Pr$=1), the C-method underestimates $Q_H$ since the air-surface temperature difference alone can not compensate for the effect of reduced momentum flux. To correct for this bias in $Q_H$, $Pr$ would need to decrease, i.e. the C-method would need to account for more effective eddy diffusivity than eddy viscosity at the given height. In the absence of wind profile measurements, we can only assume that these effects take place at our site, but we have no observational evidence for the presence of the wind speed maximum.*

● The authors need to be careful that the key results outlined in the abstract and conclusions are explicitly analysed in the paper. At present, there is some support for the alternate parameterisation schemes presented here, but they depend too much on in-situ turbulence measurements to be used widely. These results are still worth presenting, they just need to be more thoroughly analysed (perhaps elsewhere) before definitive statements can be made. It is good to see movements toward developing new turbulent heat flux parameterisations for mountain glacier environments, which is an essential step for the community.

**A valid point raised by the referee. We streamlined the conclusions to reflect more about our own data and what they show.**

● Specific Comments (page-line):
1-8: "the bulk method" – please clarify what is meant by this term. In general, the terminology used needs clarification. The term K-approach is not likely to be familiar for most readers of The Cryosphere, and is easily confused with the KInt approach introduced later. Consider using a different term here to distinguish the bulk methods in which K scales with u*, perhaps "M-O bulk schemes".

**We now clarify what is meant by the bulk methods in the abstract. The term 'K-approach' is removed, and instead we refer to a more commonly known terminology, i.e. bulk methods based on the gradient transport theory or K-theory (Stull, 1988). We also label these methods in the text as C-methods.**

● 1-12: "The OPEC-derived 30-min momentum flux is linearly related to the measured wind speed, contrary to the proposed quadratic relation by the commonly used bulk methods." – This result is not shown but rather hinted at (26-4). Needs to be clearly analysed within the paper for this statement to be supported in the abstract.

**Actually, the results have been shown (Figure 10c in the original manuscript) where u* is proportional to √(Uz), with a constant of proportionality being a function of the M-O stability parameter. This is equivalent to momentum flux being proportional to Uz, since the u* is defined as the square root of the momentum flux. Nevertheless, since this finding is not the key finding in the study, we removed it from the abstract.**

● 1-15: "In agreement with the katabatic flow model, we show that in a more stable atmosphere the bulk exchange coefficient for momentum is smaller." Again, the relationship is not analyses specifically, so it is hard to see this a key result. Please revise.

**We added more analysis on this issue in the text (dependency of the parameters in the K_Int method on the proxy variable for the environmental lapse rate). In the abstract, however, this sentence is now removed since it is not one of the key findings.**

● 1-16: "The sensible heat flux can be more successfully modeled if the bulk exchange coefficients for momentum and heat are allowed to follow different parametrization schemes, rather than assuming equal schemes as is the case in the common bulk methods." But the data don't seem to show a large improvement for the more complicated schemes when only mean wind speed and temperature are used. These schemes often rely on measurements of z/L so aren't easily transposed in space and time. Please revise.

**The conclusions here follow our findings on the Kint method (section 3.3.4 in the original manuscript):** *"The Kint -approach, however, does not perform as well for the sensible and latent heat fluxes as it does for the friction velocity. This led us to conclude that while the linear-Gaussian parametrization of K(z) works well for the momentum flux, it works poorly for the heat flux, indicating a shortcoming in the commonly used assumption that KM and KH can share the same parametrization."* **We now improved the analysis on the similarity between eddy viscosity and eddy diffusivity, and provided more discussion on the varying eddy Prandtl number.**
*Page 25 in the revised manuscript (copied here in the Latex form):*
*As already illustrated in Fig.\ \ref{fig: scatter plots with basic bulk methods}, the $\mathrm{K_{Int}}$ method gives the best estimate of the friction velocity and the poorest estimate of $Q_H$ among all the bulk methods. This performance pattern indicates that $K_M$ and $K_H$ do not necessarily share the*

*same parametrization. Setting Pr=0.75 for katabatic conditions, the $\mathrm{K_{Int}}$ method performance is simulating $Q_H$ improves (Table 3).*

● 2-15: References needed here.

**Done. We added Hock (2005) as an appropriate reference here.**

● 2-21: Please consider adding Guo, X., Yang, K., Zhao, L., Yang, W., Li, S., Zhu, M., Yao, T., and Chen, Y.: Critical Evaluation of Scalar Roughness Length Parametrizations Over a Melting Valley Glacier, Boundary-Layer Meteorology, 139, 307-332, 2011.

**Included. Thank you for the reference.**

● 3-10: "valley glaciers" – do you mean mountain glaciers? Also, it could be worth consistently referring to mountain or alpine glaciers if the two are to be treated similarly (see 4-26 & 4-27).

**Thank you for pointing out this inconsistency. Yes, we mean mountain glaciers, and have corrected this term accordingly throughout the manuscript.**

● 3-12: z0v is a mathematical variable that relates the flux and the gradient, and, as such, is not always related to the turbulence generated by roughness elements at the surface. Outer-layer turbulence can, for instance, increase the momentum flux in the surface layer, thereby changing z0v, while the surface roughness elements remain constant. Please revise.

**We agree, but in our case we refer to a turbulence generated by a friction drag. Although the roughness length for momentum is defined as a mathematical concept, it relates to the physical quantity (roughness of surface elements; Stull, 1988) when the turbulence is generated by a friction drag. We revised the text to avoid any ambiguity.**
*Page 3 in the revised manuscript (copied here in the Latex form):*
*While this 'effective' roughness length (Braithwaite, 1995) works well as a tuning parameter when the modeled turbulent fluxes are optimized to match the observed ones, it differs from its 'actual' z0v which is dependent only on the geometry and distribution of the roughness elements, assuming turbulence being generated by a friction drag.*

● 5-10: Please provide model numbers of the instruments.

**Added.**

● 6-13: Was the sonic temperature corrected for the effects of water vapour?

**Yes. The sonic temperature was corrected for humidity in the EddyPro software, using a correction based on Schotanus et al. (1983), revised in van Dijk et al. (2004) – this point appears in the revised manuscript.**

● 7-4: WPL corrects for fluctuations in the water vapour density induced by high-frequency changes air temperature, so is associated with turbulent eddies. Please revise.

**Revised. Thank you for this correction.**

● 7-28: The assumption of a melting surface, even on temperate glaciers, does not always hold during the melt season. It would be better to use the SEB, or an air temperature threshold to screen periods in which the surface is likely to be melting to validate the use of this assumption. One period in September 2012 shows air temperature well below freezing, so would almost definitely have lower surface temperature. Also, periods in late August and early September 2012 have air temperature close to 0°C, so the surface temperature is likely to be less than 0°C during these periods. Please discuss further and consider filtering bulk-method results based on periods in which high confidence can be had in the melting surface assumption.

**We agree that it would be better to measure surface temperature directly rather than using this assumption. Unfortunately, we do not have these measurements, while using the SEB closure to derive surface temperature turned out to be unreliable (mainly because the measured radiative fluxes have large errors). In particular, using the NR-Lite net radiometer sensor to estimate the outgoing longwave radiation from the measured net radiation and measured shortwave incoming and reflected radiation, turned out to be erroneous. In the evaluation of bulk methods, we did exclude the 30-min segments for which air temperature was below 1°C (now we clarify this in the text). Note that we also provide an error analysis in the calculated fluxes from the bulk methods assuming random errors in surface temperature.**
*Page 7 in the revised manuscript (copied here in the Latex form):*
*In the absence of direct measurements, the surface temperature ($T_{0}$) was assumed to be at melting point ($0^{\circ}C$) and the surface vapor pressure at saturation (6.13 hPa). The assumption of consistent melting is corroborated with the sonic ranger measurements showing persistent surface lowering throughout the observational period. To assure that the assumption holds we use only the data for which $T_{z_t} > 1^{\circ}$. In general, assuming that $T_{0}=0^{\circ}C$ works well on temperate glaciers during a melting season, and is more accurate than estimating the surface temperature from the longwave radiation measurements \citep{Fairall_etal1998} or from a SEB closure \citep{Hock2005}. Nevertheless, when the surface is not consistently melting, SEB closure can give much better results than the assumption of the melting surface (e.g. Conway and Cullen, 2013). Estimating $T_{0}$ from our radiation data, proved to be unreliable because of the poor accuracy of NR-Lite net radiometer. As part of our uncertainty analysis, we will quantify errors in our results due to the assumed rather than measured surface conditions.*

● 8-1: Assuming a melting surface in preference to outgoing longwave radiation measurements or SEB closure is predicated on there always being ample energy available for melt. Where this is not the case, SEB closure can give much better results than the assumption of the melting surface (e.g. Conway and Cullen, 2013). I agree that when the surface is most definitely melting, then making the assumption of 0°C is a good way to remove uncertainty in the calculation of surface temperature. Please revise this statement.

**The sentence is now revised to include the point that SEB closure can give much better results than the assumption of the melting surface (e.g. Conway and Cullen, 2013).**

● 8-25: Please refer to the later choice to extend the z/L range to z/L = 2 (11-13).

**Revised accordingly.**

● 10-5: It would be useful to introduce the filters used to select the 30-min periods used to compare fluxes.

**Revised accordingly.**

● 10-7: The term K-approach is introduced with no background. Is there a less ambiguous term to use for this family of bulk approaches, given that none of the acronyms include K, and you introduce KInt later as a separate method?

**We now label the common bulk method (Clog, C_M-O, etc) as C-methods. The term 'K-approach' has been removed.**

● 12-7: Why were errors associated with the air temperature not included in the error analysis?

**Our statement was initially incorrect and is now revised to read that the errors in the air-surface temperature differences are used in the error analysis (not just the errors in surface temperature).**

● 15-8: Why were low wind speed periods not included in the analysis? It would be more appropriate to only include the stationarity and wind direction filters, as the other filters are specifically designed for retrieving roughness lengths, rather than removing bad flux estimates. I would expect this to change the comparison significantly, especially the inclusion of low wind speed periods.

**Initially, we included this filter for the consistency to evaluate the bulk methods over the same filtered data used for the derivation of roughness length. In particular, we aimed to include only the 30-min segments with well-developed turbulence (u\* > 0.1 filter) in the presence of relatively strong shearing (Uz > 3 m/s). In the revised manuscript, we now include the 30-min segments with Uz > 1 m/s (but keeping the u\* > 0.1 filter). Note that the additional set of points did not alter our conclusions.**

*Page 9 in the revised manuscript (copied here in the Latex form):*
*To assure that the bulk method evaluation is performed on the high-quality measurements, all of the filters above are applied to the OPEC measured $u_*$, sensible heat ($Q_{H}$) and latent heat ($Q_{E}$), except the 'neutrality' and 'wind speed' filters. The latter two filters are modified so that that are modified: all runs with $\frac{z_{v,t}}{L}| < 2$ are included in the calculation of fluxes, as well as all runs with $U_z > 1 m s$^{1}$. The threshold of $\frac{z_{v,t}}{L}=$ 2 is chosen because the universal stability functions for stable stratification are commonly defined up to $\frac{z}{L}=$ 2 \citep{Foken2008}, which represents strongly stratified stable regime.*

● 23-16: The static stability referred to in the OG model is that of the background airmass. As we have no information on the background stability we don't know if this is necessarily reflected in the stability of the surface layer (z/L). Please revise.

**A good point and statement now corrected. We further elaborate on the issue in the results section on the K_Int method. Note that we altered our analysis of K_Int dependence on static stability by incorporating the near-surface air temperature measurements from the two stations in the glacier vicinity (AWS_up and AWS_low). This new approach now made Kint method independent of the OPEC-derived z/L.**

● 26-14: Is there a way to evaluate Kint/Kmax without measured z/L? For consistency, it would be useful to discuss if this is possible.

**Please see the response above. We have modified the way we evaluate Kint/Kmax in the revised version. Instead of using measured z/L, we evaluate K_max by its dependency on a proxy variable for background temperature lapse rate, which in our case is a difference between on-glacier and off-glacier near-surface air temperature. This proxy variable is empirically related to the strength of the katabatic flow, therefore, serving as a robust constraint for the K_max parameter.**

*Page 14 in the revised manuscript (copied here in the Latex form):*
*To adequately determine $K_{max}$ and $H_K$, detailed observations of the wind speed and temperature profiles with height are required. In the absence of these observations, and similarly to $C_kat$ method, we derive the parameter values through the optimization method that incorporates dependency of the two parameters on the proxy variable for $\gamma$ (i.e. the difference between on-glacier and off-glacier temperature). Since these two parameters ($K_{max}$ and $H_K$) are inter-related \citep{Parmhed_etal2004}, it is sufficient to optimize one parameter while keeping the other constant. The assumed constant value for $\overline{K_{max}}$= 0.8 m$^2$ s$^{-1}$ (or $\overline{H_K}$=20 m) is taken from its frequently observed values from a field study on an Icelandic glacier \citep{Parmhed_etal2004} as this was the only study we found that directly measured the two parameters. Keeping one parameter constant, the other is estimated from Eq.\ \eqref{eq:u star related to K Int} using 30-min OPEC-derived $u_*$ and log mean value of OPEC-derived $z_{0v}$ (from neutral conditions only). In this way we derive optimized $K_{max}$ (or $H_K$) value for each 30-min segment. The final step is to regress the 30-min optimized parameter values against the proxy variable. The regression is performed on the 2012 data with identified katabatic flow conditions only, while the 2010 data is used to validate the regression model.*

● 27-9: Why are those with measured z/L but not those with measured u* included? I think it would be better to only present schemes that do not use any time-varying information from the OPEC system, as these are the parameterisations that are of use to those wishing to use the bulk method.

**The methodology has now been modified, so that it addresses the bulk methods that rely on the parametrizations with the mean variables only (i.e. from AWS measurements, and not from OPEC sensor). The OPEC-derived z/L and u* are only applied in the sensitivity analysis in order to identify potential causes of discrepancies in the bulk method performance.**

*Page 9 in the revised manuscript (copied here in the Latex form):*
*In the gradient-flux relation, the eddy viscosity is parameterized as a function of $z$, $u_*$ and M-O stability parameter ($\frac{z}{L}$). Because $u_*$ and $\frac{z}{L}$, in the C-methods, are modeled rather than directly measured, any error in these modeled values can propagate into the flux estimates. Our goal in this section is to investigate the influence of the two variables, $u_*$ and $\frac{z_{v,t}}{L}$, on the bulk method performance. To do so, we estimate the turbulent fluxes from each of the four bulk schemes using the OPEC-derived $u_*$ and Obukhov length ($L$).*

● 29-15: It is unclear which filtered periods were used here. Please clarify.

**We now clearly state which filters are applied on the fluxes used in the inter-comparison. This explanation is given, as suggested by the earlier referee's comment, already in the methods sections so there is no need to reiterate it again in the discussion.**

*Page 9 in the revised manuscript (copied here in the Latex form):*
*To assure that the bulk method evaluation is performed on the high-quality measurements, all of the filters above are applied to the OPEC measured $u_*$, sensible heat ($Q_{H}$) and latent heat*

*($Q_{E}$), except the 'neutrality' and 'wind speed' filters. The latter two filters are modified so that all runs with $|\frac{z_{v,t}}{L}| < 2$ are included in the calculation of fluxes, as well as all runs with $U_z > 1$ m s$^{1}$. The threshold of $\frac{z_{v,t}}{L}=$ 2 is chosen because the universal stability functions for stable stratification are commonly defined up to $\frac{z}{L}=$ 2 \citep{Foken2008}, which represents strongly stratified stable regime.*

● 29-27: There is a need to discuss potential systematic biases in both surface and air temperature and how these could propagate into the calculation of H. This could include additional screen using the SEB to identify melting periods only to compare to OPEC fluxes, and/or a comparison of sonic-temperature with the unventilated air temperature measurements or application of corrections for low wind speed (Huwald et al. 2009) .

**As mentioned earlier, we exclude the 30-min segments that can potentially have surface temperature below 0°C (using the filter for near-surface air temperature to exceed 1°C). In addition to the assessment of errors in the modelled fluxes originating in the random errors in the surface temperature, we now also assess the errors due to the radiative overheating of the unventilated temperature sensor. To do so, we compare sonic-temperature measurements with the measurements from the naturally ventilated sensor, and derive a bias-correction function for temperature measurements.**
*Page 14 in the revised manuscript (Section 2.5 Uncertainty analysis):*
*We quantify uncertainties in the modeled 30-min fluxes due to (1) the choice of roughness lengths, (2) the assumption that the surface temperature is at a melting point, and (3) the systematic error in air temperature due to radiative heating of the temperature sensor.*

● 31-7: "fails to successfully simulate QH." I would be careful making this statement, as you could argue that it does simulate H fairly well, not just as well as some other, more site-specific schemes. The main failing of the bulk schemes presented here, is the failure to model u*. Please revise.

**Revised accordingly.**

● 31-8: "Note that the new stability correction acts in the opposite direction than those commonly used for glacier studies: in our case, the modeled QH needs to increase, rather than be suppressed, as the stability increases." Yes, but only because u* is overestimated. H still decreases relative to stable conditions. Thus, the result is not so much about the effect of atmospheric stability, but the trouble with specifying turbulence in the presence of katabatic flow. Please revise.

**Revised accordingly.**

● 33-4: This paper is not available to the reader at the present time, so it is hard to assess this statement.

**This paper is now published and we cite it accordingly.**

● 33-13 to 16: This is a key result and should feature more highly in the manuscript.

**We agree that this is one of the key findings and now highlight it more clearly.**

● 33-22: "Applying the KInt approach to assess u*, which is then used in the K-approach with the newM-O stability function to assess QH gives the best performance across all the bulk methods we tested". Yes, but the fit between u*, Uz and Kint is informed by measurements of z/L (which contains H) as is the fit between u*, z/L and H in the stability function, so it is not surprising that this function works the best. Please discuss the self-correlation and revise.

**This part of the methodology is now modified (see our responses about the K_Int method and its relation to the proxy variables for the background temperature lapse rate). In the revised version, the parameters in K_int method are calibrated independently of measured z/L (Page 14 in the revised manuscript).**

● Figures 7 and 11: Consider including the numbers assigned to each parameterisation above the columns of each figure panel to aid the reader.

**Revised accordingly.**

● Figure 12: The order of parameterisations in panel (a) needs to be consistent with Table 3 – i.e. the first parameterisations introduced at the top. As with Fig 7, needs to have the numbers assigned to the schemes next to the y-axis labels.

**Revised accordingly.**

● Table 1: Please include the units for the roughness lengths in here.

**Done as suggested.**

● Table 3: Consider removing parameterisations 17, 19, 21 and 23 as they are essentially duplicates of 16, 18, 20 and 22. Also consider adding lines between the sub-sets of parameterisations.

**We revised the table according to the new clusters of bulk methods and have excluded the duplicates.**

We sincerely thank Referee #1 for all the editorial comments. All of them have been incorporated in the revised manuscript.

**Responses to Referee #2's comments:**

● Generally the paper is well written, but I did find it cumbersome to read, because it includes many different parameterisations that are not easily distinguishable in the text. So I suggest that the authors consider a restructuring of the methods to clarify the difference between the model runs they performed and maybe "cluster" the methods that are similar.

**A great suggestion made by the reviewer. In the revised manuscript, the methods have now been clustered into two main types: C-methods and methods that rely on a katabatic model. We significantly revised the structure of the manuscript to make it more streamlined and less 'cumbersome' to read. We now feel that that paper is easier to read following this clustering and general streamlining.**

● My main concern with the paper is that it neglects the very stable conditions by only looking at conditions where wind speed is >3m/s or (the moisture/temperature gradients are large enough). I appreciate that the measurement of turbulent fluxes under very stable conditions are harder to obtain because the mean flow is non-stationary and characterised by brief episodes of intermittent turbulence Mahrt [1989]; Beljaars and Holtslag [1991]; Mahrt [1998]; Cheng et al. [2005]. Considering the significant amount of the periods where low wind speeds occur (Figure 4 in the submitted manuscript), those periods should not be neglected when trying to improve the turbulent fluxes parameterisations over glaciers. A number of studies have been dedicated to finding valid flux-profile relationships for very stable conditions, such as are often found over snow and ice surfaces [e.g. Webb,1970; Kondo et al., 1978; Lettau, 1979; Brutsaert, 1982; Holtslag and deBruin, 1988; Beljaars and Holtslag, 1991; Cheng and Brutsaert,2005; Grachev et al., 2007] and those studies have also been applied to snow and ice surfaces [Pomeroy et al., 1998; Jordan et al., 1999;Sharan, 2009; Dadic et al., 2011].

**We only use the near-neutral stability criterion in calculating the roughness lengths, while for the comparison of measured versus modeled turvulent fluxes we include all stability conditions that satisfy $-2 < z/L < 2$. In the revised manuscript we now also include the conditions for which wind speed exceeds 1 m/s (instead of the original threshold of 3 m/s). We note, however, that the inclusion of these data did not change the results of the bulk method evaluation. One needs to be careful when assuming that the very stable conditions are present during low wind speeds only → sloped glacier surfaces can have very stable conditions ($z/L > 1$) present during high wind speeds (e.g. $U_z > 5$ m/s), as we now show in Figure 7 of the revised manuscript (see below). We thank the referee for the list of references. We already have quite extensive list of references but we have added now a selection from this recommended list.**

[Figure]

*Figure 7: Modeled z/L with the fixed-point iterative scheme in the $\mathrm{C_{M-O}}$ method and the Bulk Richardson number ($R_{ib}$) against the OPEC-derived stability parameter (z/L obs). Dashed black line shows 1:1 line. Also shown is a dependency of the 30-min OPEC-derived stability parameter (z/L obs) on the wind speed ($U_z$) and the near-surface air temperature ($T_z$).*

● All bulk methods assume a logarithmic profile, and they only differ in what stability correction they use. This should be clarified in the manuscript.

**This is correct if the stability correction is negligible. However, introducing a stability function into the parametrizations for K (eddy viscosity) as K= ku\*/phi(z/L) does change the logarithmic profile in the bulk method to log-linear profile (under stable conditions). This is why we differentiate between C_log (with logarithmic profile) and C_M-O (with log-linear profile). We prefer to keep this differentiation, and clarify this issue in the revised manuscript.**

● Figures 1, 2, and some of Figure 4 (radiation, precipitation, wind direction) are not needed in this paper and can be removed.

**We prefer to keep the figures showing the study area and the location of AWS, as well as the photos of the station setup on the glacier. We also prefer to keep radiation and wind direction since we used these variables to help us identify conditions with katabatic flow. Alternatively, we could place these figures in the Supplementary material and will consult ourselves with the editor about whether this is recommended.**

● Figure 5: It is of no surprise that pretty much all 4 methods in this Figure have the same results, considering they all use he bulk method at almost neutral conditions. By neglecting the stable conditions, they don't have much reason not to vary. I am therefore not sure what the point of his comparison is.

**As noted above, we use all the stability conditions (-2 < z/L < 2), not just the near-neutral ones (-0.1 < z/L < 0.1). It seems that this was not clearly communicated in the original paper, and therefore we clarified this issue better in the revised manuscript.**
*Page 9 in the revised manuscript (copied here in the Latex form):*
*To assure that the bulk method evaluation is performed on the high-quality measurements, all of the filters above are applied to the OPEC measured $u\_*$, sensible heat ($Q_{H}$) and latent heat ($Q_{E}$), except the 'neutrality' and 'wind speed' filters. The latter two filters are modified so that all runs with $|\frac{z_{v,t}}{L}| < 2$ are included in the calculation of fluxes, as well as all runs with $U_z > 1 m s$^{1}$. The threshold of $\frac{z_{v,t}}{L}=$ 2 is chosen because the universal stability*

*functions for stable stratification are commonly defined up to $\frac{z}{L}=$ 2 \citep{Foken2008}, which represents strongly stratified stable regime.*

● P18–19: It is not surprising that the "parameterisations" which use measured u ∗ as input lead to an increase in fit with the data. u ∗ goes into the Q E equation by the power of 4, it's proportional to Q H . It changes L with the power of 3, so will disproportionally decrease z/L. Some of this discussion (why u ∗ has more influence on the turbulent fluxes calculation than z/L) might be easier to understand by just looking at the equations and the relevance of the different parameters.

**A good point. We now show the equations that relate the fluxes to u\* in the bulk method (Equation 19 and 20 in the revised manuscript). Our goal here was to test how well this relation (between the fluxes and mean variables), expected by the theory, is supported by the data. We have now moved this analysis with measured u\* and z/L into the sensitivity analysis section and have improved the discussion.**
*Page 9 in the revised manuscript (copied here in the Latex form):*
*In the gradient-flux relation, the eddy viscosity is parameterized as a function of $z$, $u_*$ and M-O stability parameter ($\frac{z}{L}$). Because $u_*$ and $\frac{z}{L}$, in the C-methods, are modeled rather than directly measured, any error in these modeled values can propagate into the flux estimates. Our goal in this section is to investigate the influence of the two variables, $u_*$ and $\frac{z_{v,t}}{L}$, on the bulk method performance. To do so, we estimate the turbulent fluxes from each of the four bulk schemes using the OPEC-derived $u_*$ and Obukhov length ($L$).*

● Furthermore, I the observation on page 18 (L1–3) that the C log and C S R methods are not justified in table 3, where the difference between the u ∗ models in the correlation coefficient r is between 0.94 and 0.95 for Q E and between 0.82 and 0.85 for Q H , which is not exactly significant. I am not sure how to address this problem, but I'm sure the authors can come up with more robust conclusions than that.

**Our discussion about the bulk method performance now reflects the results from all evaluation metrics (RMSE, MBE and correlation coefficient), not just the correlation coefficient. This section has also now been modified and moved to the sensitivity analysis section (Section 3.3.1) and the discussion has been improved, i.e. we first intercompare only the bulk methods with mean meterological variables, and later introduce the sensitivity tests when OPEC-derived u_\* and z/L are  used.**

● p 29, L1-2: Considering that the authors have most SEB components to actually calculate the surface temperature, and that the surface temperature is an important feedback for the TF, the authors should consider calculating the surface temperature and including it in their calculations using the different parametrizations. It would be interesting what effect the different parameterizations have on surface temperature. I do not expect the authors to change all their results now, but maybe it's worth a discussion in the paper.

**Using the SEB closure to derive surface temperature turned out to be unreliable, mainly because the measured radiative fluxes have large errors, in particular, the use of NR-Lite net radiometer sensor to estimate the outgoing longwave radiation from the measured net radiation and measured shortwave incoming and reflected radiation. We now explain this more clearly in the text. Also, we provide an error analysis in the calculated fluxes from the bulk methods assuming random errors in surface temperature.**
*Page 7 in the revised manuscript (copied here in the Latex form):*

*In the absence of direct measurements, the surface temperature ($T_{0}$) was assumed to be at melting point (0$^{\circ}$C) and the surface vapor pressure at saturation (6.13 hPa). The assumption of consistent melting is corroborated with the sonic ranger measurements showing persistent surface lowering throughout the observational period. To assure that the assumption holds we use only the data for which $T_{z_t} > 1^{\circ}$. In general, assuming that $T_{0}=0^{\circ}$C works well on temperate glaciers during a melting season, and is more accurate than estimating the surface temperature from the longwave radiation measurements \citep{Fairall_etal1998} or from a SEB closure \citep{Hock2005}. Nevertheless, when the surface is not consistently melting, SEB closure can give much better results than the assumption of the melting surface (e.g. Conway and Cullen, 2013). Estimating $T_{0}$ from our radiation data, proved to be unreliable because of the poor accuracy of NR-Lite net radiometer. As part of our uncertainty analysis, we will quantify errors in our results due to the assumed rather than measured surface conditions.*

● p 30, L26-30: Considering that only near-neutral conditions are used for this study, I am not surprised that the stability corrections show very little difference when modelling the fluxes.

**As already mentioned above, we use the stability conditions -2 < z/L < 2, not just the near-neutral conditions (-0.1 < z/L < 0.1). The main reason why the stability corrections did not significantly alter the fluxes is because the modelled z/L, calculated via the fixed-point iterative scheme, underestimates the OPEC-derived z/L.**
*Page 20 in the revised manuscript (copied here in the Latex form):*
*Intercomparion only across the C-methods (Fig.\ \ref{fig: scatter plots with basic bulk methods}) reveals that the performance of $\mathrm{C_{M-O}}$ method does not significantly differ from $\mathrm{C_{log}}$ method. This is because the M-O stability parameter ($\frac{z_{v,t}}{L}$), calculated with the fixed-point iterative scheme of \cite{Munro1989}, is uncorrelated with the OPEC-derived $\frac{z_{v,t}}{L}$, underestimates the stability during katabatic conditions and overestimates it during conditions with low speeds (Fig.\ \ref{fig: modeled versus observed stability}). The stability corrections that depend on calculated $\frac{z_{v,t}}{L}$, therefore have a small effect in modifying the fluxes during the katabatic conditions, while during the non-katabatic conditions with the low wind speeds the fluxes are unnecessarily suppressed. Furthermore, we found no correlation between the 30-min OPEC-derived $\frac{z_{v,t}}{L}$ and any of the mean meteorological variables (e.g. temperature, wind speed; Fig.\ \ref{fig: modeled versus observed stability}), which explains the failure of the fixed-point iterative scheme that relies on these dependencies. The poor performance of the stability corrections in $\mathrm{C_{Rib}}$ method also follows from the lack of correlation between $R_{ib}$ and the OPEC-derived $\frac{z_{v,t}}{L}$ (Fig.\ \ref{fig: modeled versus observed stability}).*

● P31, L11-13: As far as I remember, the reason why the turbulent fluxes are suppressed in Conway and Cullen (2013) is that they assumed the log-linear relationship to be valid under very stable conditions. The log-linear relations, however, do not allow for significant fluxes to occur at very strong stability[Monin and Yaglom, 1971; Mahrt, 1998; Pleim, 2006] and underestimate the turbulent fluxes over these conditions [e.g. Deardorff, 1968; Webb, 1970; Kondo et al., 1978; Louis, 1979; Hogstrom, 1988; Launiainen, 1995; Mahrt, 1998; Jordan et al., 1999; Stossel et al., 2010].

**Yes, this log-linear relationship explains part of the story in the findings from Conway and Cullen (2013). The other part is related to the presence of low wind maximum height. We have now incorporated this explanation more clearly in our discussion. We thank the referee for the provided references, a selection of which we included in the revised manuscript.**
*Page 22/23 in the revised manuscript (copied here in the Latex form):*

*While the original C-methods, however, overestimate $Q_H$ during the katabatic conditions, the C-methods with stability corrections and measured $u_*$ underestimate the fluxes (Table \ref{tab:evaluation results} and Fig.\ \ref{fig: scatter plot with sensitivity tests}). The overestimation of $Q_H$ during katabatic flows has also been shown in \cite{Denby_Greuell2000} and explained by a failure of M-O theory in the presence of shallow katabatic wind speed maximum. At the wind speed maximum, measured $u_*$ approaches zero, while the C-method assumes constant momentum flux in the surface layer and therefore overestimates $u_*$. The overestimation is less pronounced for $Q_H$ than for $u_*$ because the reduced turbulence at the wind speed maximum leads to an increase in the air-surface temperature difference, and subsequently an increase in the measured $Q_H$. However, when measured $u_*$ is used in the $\mathrm{C_{M-O}}$ or $\mathrm{C_{Rib}}$ method, assuming that the eddy diffusivity is as effective as eddy viscosity ($Pr$=1), 
[revised manuscript text omitted]

---

## Author Response (AR2)

**Responses to reviewers' comments on the revised manuscript**

Dear Dr. Thomas Mölg,

Please find attached our responses to the reviewer's comments and the revised manuscript, including the revised manuscript with tracked changes. We thank you for handling the manuscript and hope the revised version meets the standards for a publication in TC.

Best regards,
Valentina Radić

**Responses to reviewer**

*General Comments*
*The authors have thoroughly addressed the comments raised in my initial review and present a much-improved manuscript. The new and revised analyses make the results much more compelling and present a clear message. In particular, the calculation of Kmax using on-off glacier temperature difference is an addition aspect of novelty and a significant contribution. The inability of measurement errors to account for the observed differences between the model and measurements adds additional urgency to the need for further work to derive better functions to compute turbulent heat fluxes over glaciers. The combination of the two best performing methods for friction velocity and the heat fluxes provides a new avenue for further investigation. I have some remaining minor comments, but no overall general concerns. The manuscript presents a well-constructed and valuable contribution to our understanding of turbulent heat fluxes over glacier surfaces.*

We thank the reviewer for his support of this study and for a detailed review on the initial version of this manuscript.

*Specific Comments (page-line):*

*24:2-6 The log-log correlation between observed k and the on-off glacier temperature difference is not very strong (r = -0.34) and the figure presented (second two panels of Figure 10) doesnt support the statement that as the temperature difference increases, indicative of higher static stability, the k parameter decreases. From the figure it seems as if high values of k dont appear under large temperature differences, but certainly there is no obvious power-law relationship presented. A log-log plot may highlight this feature. This result is not central to the main result, so I would suggest revising the text or removing this analysis.*

We thank the reviewer for catching this spurious result. Indeed, the linear regression in the log-log plot is not resistant to outliers and, in our analysis, the significant correlation (r = -0.34) is due to the outliers. Since this result is not central to our study we removed the last two panels in Figure 10, while in the text we now mention that no significant correlation was found between 'observed k' and on-off glacier temperature difference.

*25:9 It is a little ambiguous in the text what formulae were used to calculate heat fluxes for the hybrid method. From what I can tell, equations 19-20, 22-23 were used to calculate $Q_H$ and $Q_E$, but that instead of calculating $u_*$ with equation 16 and 21, $u_*$ is calculated with equations 33 and 36. Is this correct? If so, then perhaps the full list of equations can be given. The text at 25:12 should also refer to exchange coefficients for heat rather than heat fluxes. Also, please describe how $K_{Int}$ was calculated for the hybrid method (presumably from equation 38?) as well as the value for $Pr$ used.*

This is correct. We modified the text to clarify these equations:

'This variant of the bulk method calculates the friction velocity from the $K_{Int}$ method (Eq. (33) and (36)), which is then inserted in the $C_{log}$ method's equations for $Q_H$ and $Q_E$ (Eq. (19) and (20)) with $Pr=1$.'

*Editorial Comments*

*23:2 Please change the use of parenthesis to denote opposites into a full sentence. E.g. explains 61 % of the variance in the 2012 observations and 19 % of variance in 2010 observations.*

Done

*25:14 Please add reference to Table 2 with Figure 8. It is sometimes difficult to understand which panel of a figure is being referred to in the text and to match the figure captions describing each panel. It would be useful if the multi-panel figure panels were labelled (a), (b), (c)... etc. to aid the reader.*

Table 2 is now referenced. The suggested labeling is now added to all multi-panel figures except for figures 4 and 8, where a title is given for each panel showing the same variables on x and y axis.

[revised manuscript text omitted]